# Comprehensive mapping of SARS-CoV-2 interactions in vivo reveals functional virus-host interactions

Siwy Ling Yang[1,8], Louis DeFalco [2,8], Danielle E. Anderson [3,8], Yu Zhang [1,8], Jong Ghut Ashley Aw [1,8], Su Ying Lim[1], Xin Ni Lim[1], Kiat Yee Tan[1], Tong Zhang[1], Tanu Chawla[3], Yan Su[4], Alexander Lezhava [4], Andres Merits[5], Lin-Fa Wang [3,9 ✉], Roland G. Huber [2,9 ✉] & Yue Wan [1,6,7,9 ✉]

SARS-CoV-2 is a major threat to global health. Here, we investigate the RNA structure and RNA-RNA interactions of wildtype (WT) and a mutant (Δ382) SARS-CoV-2 in cells using Illumina and Nanopore platforms. We identify twelve potentially functional structural elements within the SARS-CoV-2 genome, observe that subgenomic RNAs can form different structures, and that WT and Δ382 virus genomes fold differently. Proximity ligation sequencing identify hundreds of RNA-RNA interactions within the virus genome and between the virus and host RNAs. SARS-CoV-2 genome binds strongly to mitochondrial and small nucleolar RNAs and is extensively 2'-O-methylated. 2'-O-methylation sites are enriched in viral untranslated regions, associated with increased virus pair-wise interactions, and are decreased in host mRNAs upon virus infection, suggesting that the virus sequesters methylation machinery from host RNAs towards its genome. These studies deepen our understanding of the molecular and cellular basis of SARS-CoV-2 pathogenicity and provide a platform for targeted therapy.

[1] Epigenetic and Epitranscriptomic Regulation, Genome Institute of Singapore, Agency for Science, Technology and Research (A*STAR), Singapore, Singapore. [2] Biomolecular Function Discovery, Bioinformatics Institute (BII), Agency for Science, Technology and Research (A*STAR), Matrix #07-01, Singapore, Singapore. [3] Programme in Emerging Infectious Diseases, Duke-NUS Medical School, Singapore, Singapore. [4] Laboratory of translational diagnostics, Genome Institute of Singapore, Agency for Science, Technology and Research (A*STAR), Singapore, Singapore. [5] Institute of Technology, University of Tartu, Tartu, Estonia. [6] School of Biological Sciences, Nanyang Technological University, Singapore, Singapore. [7] Department of Biochemistry, Yong Loo Lin School of Medicine, National University of Singapore, Singapore, Singapore. [8] These authors contributed equally: Siwy Ling Yang, Louis DeFalco, Danielle E. Anderson, Yu Zhang, Ashley J Aw [9] These authors jointly supervised this work: Lin-Fa Wang, Roland G. Huber, Yue Wan ✉email: linfa.wang@duke-nus.edu.sg; rghuber@bii.a-star.edu.sg; wany@gis.a-star.edu.sg

Coronaviruses (CoVs) are enveloped viruses with positive-sense single-stranded RNA genomes. They are widespread in animals and can cause mild to severe respiratory or enteric disease in humans[1,2]. There are currently seven CoVs known to infect humans, which include the four "seasonal" human CoVs: OC43, 229E, NL63, and HKU1[3], which can cause mild cold-like symptoms and three highly pathogenic CoVs: SARS-CoV, MERS-CoV, and SARS-CoV-2. SARS-CoV emerged in 2002 and resulted in more than 8,000 human infections with a case fatality rate of approximately 10%[4]. This was followed by the discovery of MERS-CoV in 2012, which has resulted in more than 2000 human infections and over 800 lethal cases with ongoing sporadic outbreaks in the Middle East[5]. More recently, the outbreak of SARS-CoV-2 has caused unprecedented social and economic damage around the world[6]. Since it was first reported in Wuhan, China in December 2019, SARS-CoV-2 has resulted in over 164 million infections and more than 3.4 million deaths as of 20th May 2021, according to WHO. Different variants of SARS-CoV-2 have been found to circulate within patients. Viruses that contain deletions of various sizes in the ORF8 region have been found around the world, including in Singapore, Taiwan, Bangladesh, Australia, and Spain[7]. In particular, a 382-nucleotide deletion (Δ382) of the SARS-CoV-2 genome that truncates ORF7 and deletes ORF8 was found in patients in Singapore[8]. While patients infected with Δ382 virus showed less severe symptoms than those infected with wild-type (WT) viruses, the molecular mechanisms behind virus attenuation in patients are unclear[7]. It is hence imperative to understand how SARS-CoV-2 and their variants function, in order to facilitate effective surveillance, prevention, and treatment strategies.

CoV genomes are among the largest of the RNA viruses, with lengths of 26–32 kb[2]. Upon entry into the cell, the positive-sense genome is translated from two open reading frames (ORF1a and ORF1ab) and the resulting polyproteins are cleaved into non-structural proteins. Non-structural proteins are essential for virus RNA replication that, in addition to new genomes, also generate numerous subgenomic RNA (sgRNA) species using discontinuous transcription[9]. These RNAs, together with the full genome, can interact with host cell proteins and RNAs to regulate virus infection. Like many other RNA viruses such as dengue virus (DENV) and Zika virus (ZIKV), the SARS-CoV-2 genomic RNA can fold into secondary and tertiary structures that are essential for virus RNA replication and protein translation[10–13]. Importantly, elements in the 5' and 3' untranslated regions (UTRs) have been implicated in virus replication and protein synthesis[14,15], and the frameshifting element is important for ribosome slippage to enable the translation of ORF1ab[16,17]. However, how the rest of the virus genomes folds into short- and long-range structures is still under-studied.

Here, we utilize different high throughput RNA and interactome techniques (SHAPE-MaP[18], PORE-cupine[19], and SPLASH[20]) to comprehensively interrogate the secondary structures and virus-host interactions along with the WT and Δ382 SARS-CoV-2 genomes to identify potentially functional structure elements along the virus genome (Fig. 1a). Using PORE-cupine, we identify sgRNA-specific structures as well as WT and Δ382 specific structures using Nanopore direct RNA sequencing. The advantage of long-read sequencing enables us to map our sequencing reads uniquely to each sgRNA, without needing to average structure signals across all sgRNAs and full-length RNA in short-read sequencing, due to ambiguous mapping. SPLASH further allows us to identify pair-wise RNA interactions using proximity ligation and sequencing, deepening our knowledge of how the genome folds along itself for function. In addition to studying how the virus genome interacts with itself, determining how the virus genome interacts with host RNAs in its cellular

environment is another key to understanding virus pathogenicity. Other RNA viruses, including ZIKV, have been shown to directly interact with host RNAs such as microRNAs to impact virus infection[11]. Most host factor studies for SARS-CoV-2 done to date have been focused on how the host proteins interact with the virus proteins and genome; much less is known about how SARS-CoV-2 interacts with host RNAs inside cells[21–24]. Here, we utilize proximity ligation sequencing (SPLASH) to identify host RNAs that interact with the SARS-CoV-2 genome inside infected Vero-E6 cells (Fig. 1a). We observe that SARS-CoV-2 RNA interacts strongly with a small nucleolar RNA (snoRNA) SNORD27 and is 2'-O-methylated inside cells. We further show that 2'-O-methylation of host RNAs is decreased in SARS-CoV-2 infected cells, and that virus-SNORD27 interaction could serve as a mechanism for SARS-CoV-2 to facilitate host RNA degradation.

## Results

**SARS-CoV-2 RNA is highly structured in host cells.** To study the secondary structures of SARS-CoV-2 RNAs inside cells, we infected Vero-E6 cells with WT and Δ382 SARS-CoV-2 and performed structure probing using the compound NAI (Methods)[10,25]. We then performed mutational mapping (MaP) to determine the location of high reactivity bases, indicating single-stranded bases, along the virus genome. We confirmed that mutational rates along with NAI-treated, denatured and DMSO-treated samples are as expected (Supplementary Fig. 1a,b, Supplementary Data 1). Biological replicates of SARS-CoV-2 SHAPE-MaP show that the reactivities across replicates are highly reproducible (Supplementary Fig. 1c), cover around 80% of the entire SARS-CoV-2 genome (Fig. 1b), and map to known structures in the 5' and 3' UTR as expected (Supplementary Fig. 1d, Supplementary Data 2). SHAPE-MaP reactivities were used to constrain RNA secondary structure predictions to obtain accurate structure models of the entire SARS-CoV-2 genome (Methods, Fig. 1c,d, Fig. 2)[26]. Our structure models are consistent with previously identified structural elements in the 5' and 3' UTRs (Fig. 1c)[27–29] and for TRS-L elements (Supplementary Fig. 2a)[27], confirming that models based on our data are accurate. As the frameshifting element is a conserved element that is important for ribosome frameshifting and translation of ORF1b, multiple structure models based on high throughput structure probing data have been proposed. We observed that our structure model resembles the alternative structure presented in Lan et al. (Fig. 1d, Supplementary Fig. 2b), and mapped our SHAPE-MaP reactivities onto the different proposed models in the literature. Out of five different structures, our SHAPE-MaP reactivities (from both the WT and Δ382 SARS-CoV-2 virus) agree the most with the in-cell FSE model proposed by Lan et al. (Supplementary Fig. 3,4). Similar to other RNA viruses like DENV and ZIKV, 57% of the bases in the SARS-CoV-2 genome are predicted to be paired, with a median helix length of 5 bases in both WT and Δ382 genomes (Supplementary Fig. 4b)[10]. These short helices enable RNA viruses to escape from host immune responses.

To identify potentially functional structural RNA elements in the SARS-CoV-2 genome, we used a consensus model between WT and Δ382, incorporating local SHAPE reactivity, local Shannon entropy of the structure models and local ScanFold Z-scores in 150 nt windows (Fig. 2a)[29–32]. We evaluated window sizes of 50–300 nt which yielded consistent results (Supplementary Fig. 6a). We considered a position a consensus candidate if it had at least four out of six possible characteristics of 'low average SHAPE' as an indication of structuredness at a location, 'low Shannon entropy' as an indication of structural consistency and limited alternative folding, and 'low ScanFold Z-score' as a proxy for the high stability of putative structural elements in both WT

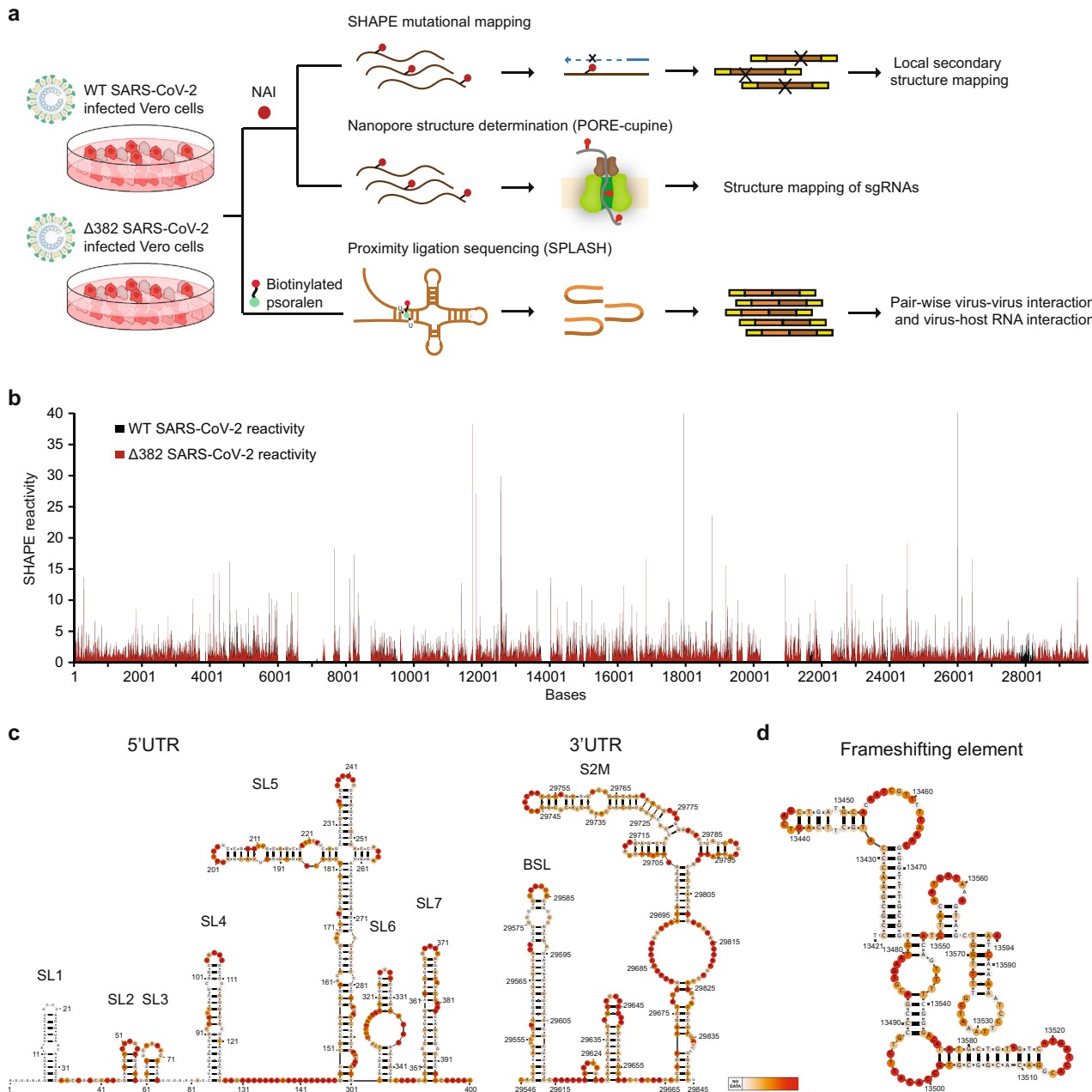

**Fig. 1 Comprehensive structure and interactome mapping of SARS-CoV-2 RNA in cell. a** Schematic of the different strategies that were used to probe WT and Δ382 inside infected Vero-E6 cells. NAI was used to modify single-stranded bases in cells and these modifications are then either read out directly using Nanopore direct RNA sequencing or converted into a cDNA library for Illumina sequencing. Biotinylated psoralen was used to crosslink pair-wise RNA interactions in infected cells to capture both intramolecular and intermolecular RNA-RNA interactions. **b** SHAPE-reactivity along with the WT (black) and Δ382 (red) SARS-CoV-2 genome. Higher reactivity regions tend to be more single-stranded. **c, d** Structure models are generated using the program RNA structure using SHAPE-reactivities as constraints[54], and visualized using VARNA[57]. The modelled structures of 5′ UTR and 3′ UTR (**c**), and that of the frameshift element (**d**) agree with known models in the literature. The frameshift element model is the highest confidence structure generated by RNA structure using SHAPE-MaP as constraints. Source data are provided as a Source Data file.

and Δ382 (Fig. 2a). Local structure models of consensus regions were largely consistent with structures obtained in the global context and identified novel highly structured elements within the genome (Fig. 2b, Supplementary Fig. 5). As single-stranded regions present in the SARS-CoV-2 genome could be used for siRNA targeting, we also identified locations with high reactivities (top 20%) in both the WT and Δ382 genomes (Supplementary Fig. 6b, Supplementary Data 3). We identified a total of 21 regions that could be used for siRNA targeting, to facilitate potential treatments for COVID-19 disease.

**SARS-CoV-2 genome contains hundreds of regions involved in intramolecular long-range interactions**. In addition to determining which bases are paired or unpaired in the SARS-CoV-2 genome, we also wanted to know the identity of pairing partners within the genome. To identify pair-wise RNA interactions, we treated SARS-CoV-2 or Δ382 infected Vero-E6 cells with biotinylated psoralen and performed proximity ligation sequencing using SPLASH[20]. Biological replicates of SPLASH showed a good correlation of pair-wise interactions between the samples, indicating that our method is robust (Supplementary Fig. 7a,b). 82.9%

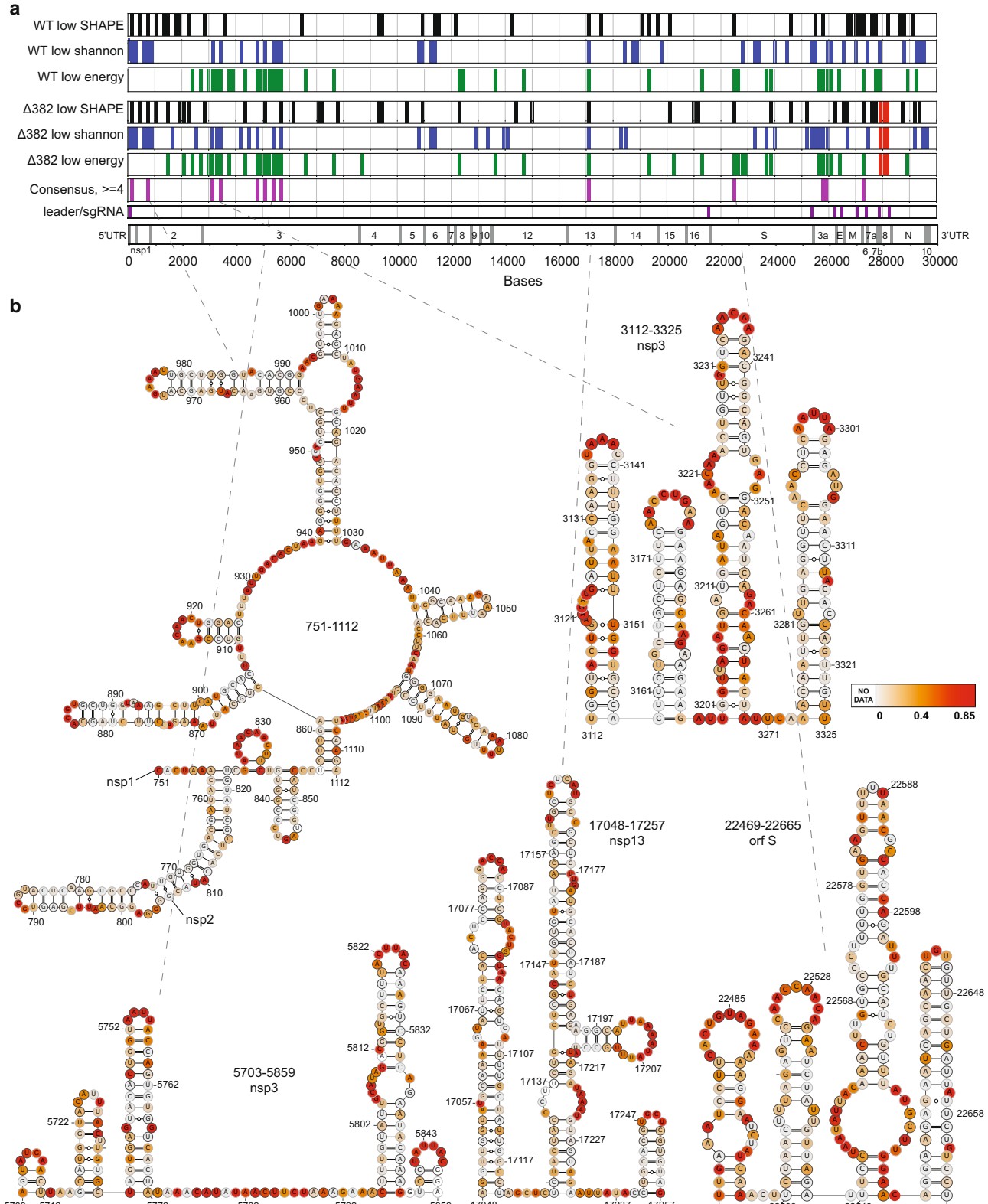

**Fig. 2 SHAPE-MaP identifies functional structural elements along the SARS-CoV-2 genome. a** Regions (150 bases) with the lowest 20% SHAPE reactivity (in black), Shannon entropy (blue), and greatest difference (top 20%) between actual and shuffled energies (green) are shown along both the WT and Δ382 genomes. 12 consensus regions are consistently highlighted (4/6) across both WT and Δ382 genomes and are shown in fuchsia. The red box indicates the deletion region in the Δ382 genome. **b** Structure models of 5 consensus were generated using the program RNAstructure[54], using SHAPE-MaP reactivities as constraints, and visualized using VARNA[57]. The SHAPE-reactivities are mapped onto the structure models. Source data are provided as a Source Data file.

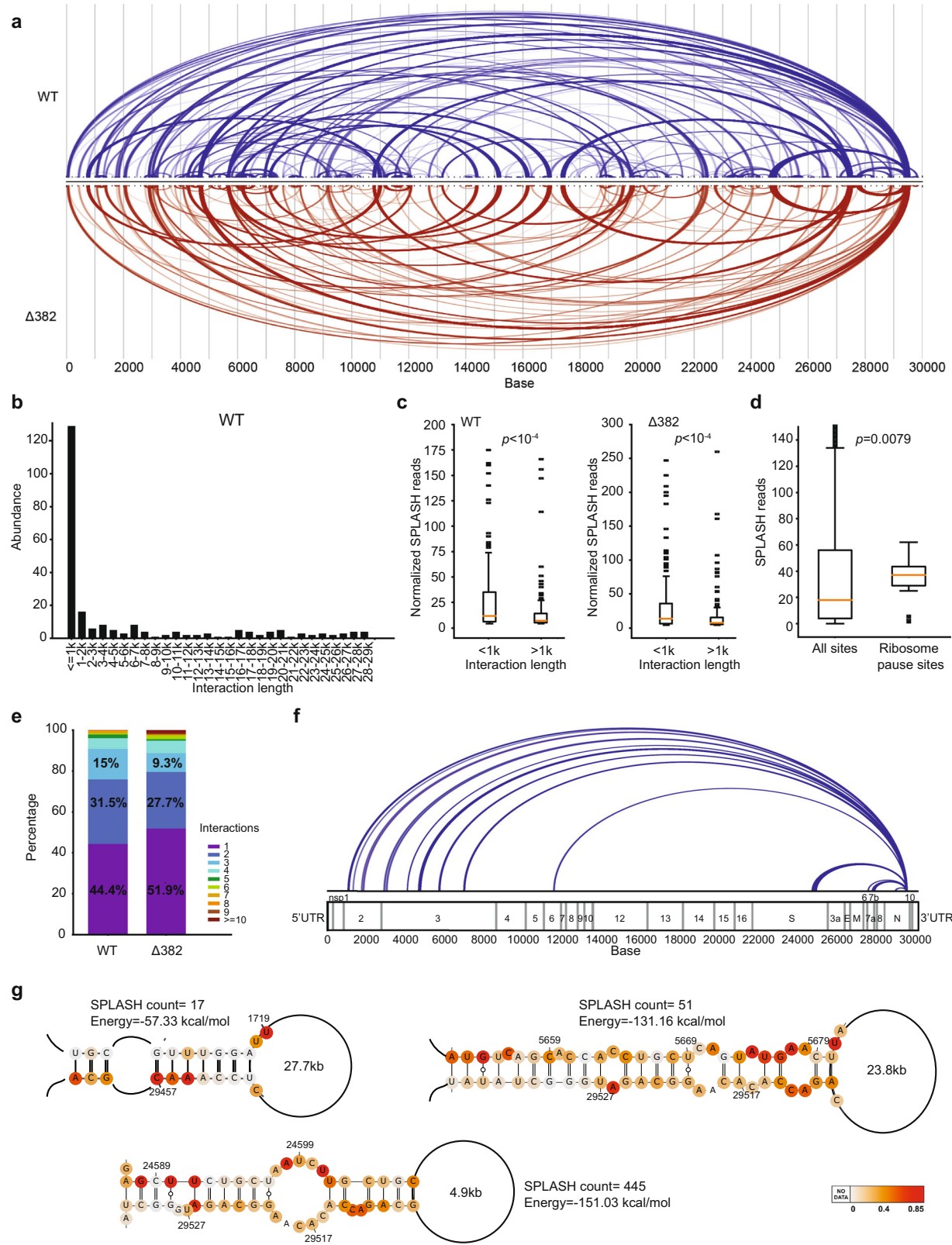

of our chimeric interactions on the 18 S and 28 S rRNA fall within 30 Å in physical space, indicating that our data is capturing pair-wise interactions as expected (Supplementary Fig. 7c).

We identified 237 and 187 intramolecular interactions along the WT and Δ382 genomes, respectively (Fig. 3a, Supplementary Data 4,5). SPLASH pair-wise interaction patterns are largely consistent between WT and Δ382 genomes, indicating the

robustness of our method (Fig. 3a, Supplementary Fig. 7d, e). 45.6% and 42.3% of the intramolecular interactions occur over a distance longer than 1 kb in the WT and Δ382 genomes respectively, indicating that the viral sequences are involved in extensive long-range interactions (Fig. 3b, Supplementary Fig. 7f–h). Longer-range interactions (>1 kb) tend to have a lower number of reads than shorter-range interactions, indicating

**Fig. 3 SARS-CoV-2 contains hundreds of intramolecular long-range interactions. a** Pair-wise RNA-RNA interactions along the WT (blue) and Δ382 (red) genomes. The thickness of the lines indicates the abundance of chimeric reads for that particular interaction. **b** Histogram showing the distribution of interactions that span different lengths along the WT SARS-CoV-2 genome. Interactions over a distance longer than 1 kb are classified as "long-range" and comprise 45.6% of all interactions. **c** Boxplot showing the distribution of the abundance of SPLASH chimeric reads for long (>1 kb) (WT $n = 195$, Δ382 $n = 264$) and short (≤1 kb) (WT $n = 140$, Δ382 $n = 150$) pair-wise interactions in both WT (left) and Δ382 (right) genomes. Long interactions tend to have lower SPLASH interaction counts, suggesting that they may be formed more transiently. $P$-values were calculated using a two-sided Wilcoxon Rank Sum test without adjustments ($p = 3.857 \times 10^{-5}$ and $1.776 \times 10^{-8}$ respectively). **d** Boxplot showing the distribution of SPLASH chimeric reads along the SARS-CoV-2 genome for all sites (left) ($n = 29,847$) and for sites that show ribosome pausing events (right) ($n = 80$). Sites with ribosome pausing events show higher SPLASH chimeric reads, indicating that they reside in more highly structured regions. $P$-value was calculated using the two-sided Wilcoxon Rank Sum test without adjustments. In (**c, d**), the box represents the 25–75th percentiles, and the median is indicated. The whiskers show the minimum and maximum values. The outliers are presented as dots. **e** Bar-charts showing the proportion of unique pair-wise interactions, as well as interactions that have 2 or more alternative partners, along the WT and Δ382 genomes. **f** Arc plots showing the alternative interactions between N and other positions along the SARS-CoV-2 genome. **g** Representative structure models for interactions between N and other regions along the genome. Structure models are generated using RNAcofold from regions identified by SPLASH. For each interaction, the SPLASH count and predicted energy of folding from RNAcofold is shown next to the model[35]. SHAPE-MaP reactivities are mapped onto the bases in the structure models. Source data are provided as a Source Data file.

that they are formed more transiently inside cells (Fig. 3c), consistent with previous literature that longer-range interactions tend to be disrupted[33]. As RNA structures could have an impact on the regulation of virus gene expression, we examined whether RNA pairing could be associated with translation using publicly available SARS-CoV-2 ribosome profiling data (Supplementary Data 6). We observed that ribosome pause sites from cycloheximide experiments have more pair-wise interactions than non-pause sites (Fig. 3d)[34], suggesting that RNA structures could be associated with translational pauses and thus regulate the translation of SARS-CoV-2.

Interestingly, we observed that SARS-CoV-2 RNA exhibit more alternative interactions than DENV and ZIKV RNAs inside the cell, with 55.6% and 48.1% of the WT and Δ382 pair-wise interactions involving two or more partners (Fig. 3e)[10]. This suggests that SARS-CoV-2 RNA takes on numerous conformations that are present simultaneously inside host cells. We observed that a location at the 3' end of sgRNA N is particularly promiscuous and interacts with regions throughout ORF1a (Fig. 3f). Structure modelling of SPLASH identified interactions using the program RNAcofold revealed that energies calculated from the predicted pairings are coherent with the SPLASH interaction counts (Fig. 3g)[35], indicating that the relative abundance of SPLASH counts between different interactions could serve as a proxy for the relative prevalence of these interactions inside cells.

**SARS-CoV-2 sgRNAs are structurally different.** In addition to the synthesis of the full-length genomes, a nested set of 3' co-terminal sgRNAs are made in SARS-CoV-2 infected cells using discontinuous RNA synthesis[9] (Supplementary Fig. 8a). These sgRNAs range from 2–8 kb long, contain a leader sequence and are produced at different amounts. While SHAPE-MaP provides information on single nucleotide SHAPE along the genome, short-read sequencing makes it difficult to map structure information unambiguously to individual sgRNAs. While sgRNAs have been observed to have different structures from full-length genomic RNA using enrichments and proximity ligation and sequencing[36], it is unclear whether each individual sgRNA contains unique structures that could be important for sgRNA-specific functions and regulation.

To address this, we utilize our previously developed method of coupling RNA structure probing with Nanopore direct-RNA sequencing (PORE-cupine) to allow us to read out SHAPE reactivities along long RNA molecules[19]. Sequencing of two biological replicates of RNAs extracted from NAI-treated, WT and Δ382 SARS-CoV-2 infected Vero-E6 cells showed good structure correlation, indicating that our data is reliable

(Supplementary Fig. 8b, c). We also confirmed that PORE-cupine reactivity shows a good correlation with SHAPE-MaP reactivity along the SARS-CoV-2 genome (Supplementary Fig. 8d). By filtering for the full-length reads that contain leader sequences, we determined reactivities along individual ORF3a, E, M, ORF6, ORF7a, ORF7b, ORF8 (WT only), and N sgRNAs (Fig. 4a, Supplementary Fig. 9a, Supplementary Data 7). We observed that ORF7b sgRNA contains the highest average reactivities for both WT and Δ382, suggesting that it is likely to be the most single-stranded among the different sgRNAs of SARS-CoV-2 (Fig. 4b, Supplementary Fig. 9b). As structures around the leader sequences for each sgRNA were previously shown to have weak correlations with gene expression, we calculated the correlation between PORE-cupine reactivity around TRS-B sites for each sgRNA and their relative abundance from our Nanopore data. We observe a weak positive correlation between reactivity and transcript abundance, similar to previously published literature[28], for both WT and Δ382 sgRNAs, suggesting that single-strandedness around the TRS-B region could result in increased synthesis of corresponding sgRNAs (Fig. 4c, Supplementary Fig. 9c).

To identify structures unique to each sgRNA, we compared the reactivities among individual sgRNAs to identify highly consistent as well as divergent structural regions (Fig. 4a, Methods). We found 4 regions in RNAs of WT SARS-CoV-2 that showed consistent structure differences between different sgRNAs, 3 of which are also seen in the RNAs of Δ382 sgRNAs (Supplementary Fig. 9a). While two regions centred around bases 27,800 and 28,250 correspond to the leader sequences of sgRNAs of ORF7b and N respectively, two other structurally different regions (centred around 29,300 and in 3' UTR) are present within all sgRNAs, and hence cannot be identified using short-read sequencing (Fig. 4d, e, Supplementary Fig. 9a, d, e). We checked that the regions that show diverse structures in different sgRNAs also exhibit multiple interaction partners by SPLASH, confirming that those regions do exist in alternative conformations (Fig. 4f). We then visualized the sgRNA-specific structures by incorporating PORE-cupine reactivities into structure modelling and observed different structure models for the same sequence region in different sgRNAs (Fig. 4g, Supplementary Fig. 10a, b), further confirming that different sgRNAs could exist in different structures despite sharing the same sequences.

**Genomes of WT and Δ382 SARS-CoV-2 contain different RNA structures.** Viruses that contain genomes with various ORF8 deletions have been found in patients around the world[7], however the mechanisms behind how such deletions impact the virus are still largely unknown. To determine whether virus phenotypes

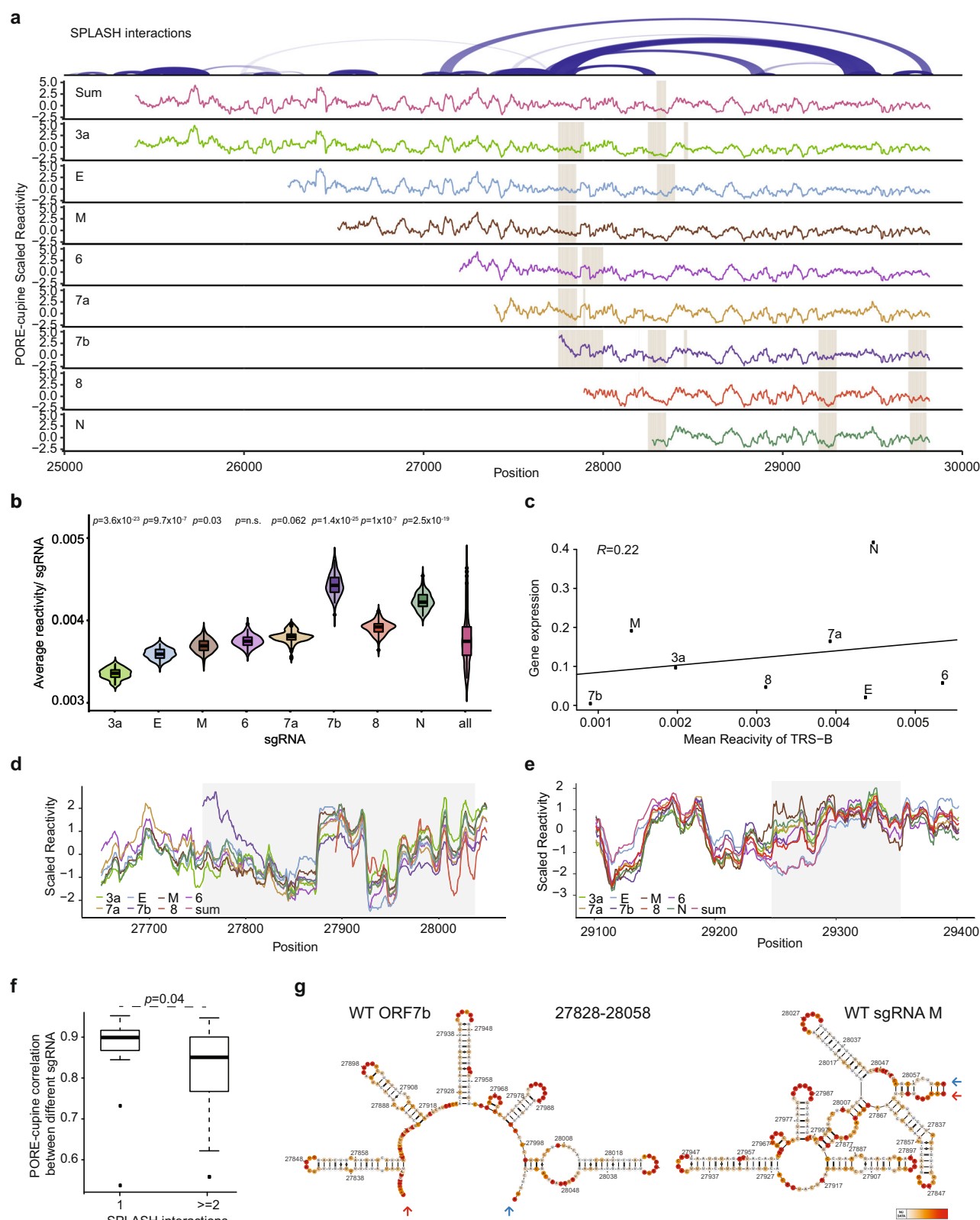

could be associated with structural differences, we performed correlations of SHAPE-MaP reactivities between the two genomes. As expected, structures in WT and Δ382 genomes are generally highly correlated ($R = 0.62$, Supplementary Fig. 11a), although we do observe local structure differences at the deletion region of around base 28000 (Supplementary Fig. 11b, c). SPLASH analysis around the deletion region also revealed

differences in pair-wise interactions between WT and Δ382, confirming the local structure rearrangements between the two viruses (Supplementary Fig. 11d).

As the deletion occurs around base 28,000 (ORF8), it is present not only in the full-length genome but also in most of the sgRNAs (except for sgRNA of N, starts the site of which is located downstream of the deletion). Due to the extensive amount of

**Fig. 4 PORE-cupine reveals sgRNA-specific structures. a** *Top*, SPLASH interactions along SARS-CoV-2 from the region 3a to the 3' UTR. *Bottom*, PORE-cupine reactivity signals are averaged across all the signals from the sgRNAs (Sum). PORE-cupine reactivity signals are also shown for 3a (green), E (blue), M (brown), 6 (purple), 7a (light brown), 7b (navy), 8 (red) and N (dark green). PORE-cupine reactivity signals for each sgRNA are filtered for full-length sequences that contain leader sequences for each sgRNA. Regions with significant differences are highlighted in grey, $p < 0.05$, (Methods). **b** Violin plots showing the distribution of average reactivities for each sgRNA. Each sgRNA is subsampled for 500 strands before calculating its mean, $n = 100$. P-values were calculated by comparing the distribution of the reactivities in each sgRNA against all of the sgRNAs with a two-sided Wilcoxon Rank Sum test. n.s.: not statistically significant. The box represents the 25–75th percentiles, and the median is indicated. The whiskers show the minimum and maximum values. **c** Scatterplot showing the correlation between the PORE-cupine reactivity around TRS-B for each sgRNA (x-axis) against transcript levels inside cells (y-axis). **d,e** Reactivity plots of regions that show significant structure differences between the sgRNAs. **d** P-value for the follow regions are: 27,750–27,850 (p-value = $3.52 \times 10^{-6}$), 27,800–27,900 (p-value = $2.55 \times 10^{-8}$), 27,850-27,950 (p-value = 0.02), 27,900–28,000 (p-value = 0.03) and 27,950–28,050 (p-value = 0.04). **e** P-value for the follow regions are: 29,250–29,350 (p-value = 0.02). **f** Boxplots showing the distribution of correlation between reactivities of different sgRNAs for regions that show unique SPLASH interactions (1) and regions that show alternative SPLASH interactions (≥ 2). Regions that show alternative SPLASH interactions take on different conformations and show lower reactivity correlations between sgRNAs. The p-value was calculated using a two-tailed Wilcoxon Rank Sum test. The box represents the 25–75th percentiles, and the median is indicated. The whiskers show the minimum and maximum values. The outliers are presented as dots. **g** Structure models of WT ORF7b and M sgRNA are generated using the program RNA structure, using PORE-cupine reactivities as constraints. PORE-cupine reactivities are mapped onto the secondary structure models. The red and blue arrows indicate the same positions (start for red and end for blue) in the structure models. Source data are provided as a Source Data file.

sequence similarity between the different sgRNAs, it is difficult to map uniquely to each individual sgRNA using short-read sequencing. Consequently, it remained unclear whether the structure differences between WT and Δ382 are present in all the sgRNAs or only between some specific sgRNAs. To determine which sgRNA shows reactivity differences between WT and Δ382 genomes, we compared the PORE-cupine reactivity profiles of individual WT and Δ382 sgRNA to each other (Supplementary Fig. 12a). While we could only detect very local reactivity differences immediately before and after the deletion site when all the WT and Δ382 sgRNA reads are summed and compared to each other as an aggregate (similar reactivity profiles obtained using short-read Illumina sequencing), we observed additional structure differences when individual WT and Δ382 sgRNAs are compared to each other (Supplementary Fig. 12a–d). We observed the largest structure differences between WT and Δ382 in ORF3a and E sgRNAs (Supplementary Fig. 12a). We also consistently observed a second structurally different region between WT and Δ382 sgRNAs at the bases 29,200–29,400 (Supplementary Fig. 12b, d), indicating that the deletion could impact distal structures that are located more than 1 kb away. As expected, we did not observe reactivity differences between N-gene sgRNAs of WT and Δ382 viruses as this sgRNA is transcribed using TRS located downstream of the deletion region. This finding indicates that the reactivity differences between other sgRNAs of WT and Δ382 viruses are likely to occur in cis due to the deletion and not due to factors that may act in trans (Supplementary Fig. 12a, b). As sgRNA of N is by far the most abundant sgRNA of SARS-CoV-2 and it did not show structure differences between WT and Δ382 viruses[9], differences in the reactivity between the 29,300 region in WT and Δ382 genomes were masked when an aggregate reactivity of all sgRNAs is used for comparison (Supplementary Fig. 12a, b). As such, using long-read sequencing to map RNA structures across sgRNAs can yield novel insights into sgRNA-specific RNA structures.

**SARS-CoV-2 genome interacts strongly with mitochondrial RNAs and snoRNAs.** The genomes of RNA viruses can interact directly with host RNAs to facilitate or restrict viral infection. By analysing the SPLASH interactions between SARS-CoV-2 and host cell RNAs, we identified 374 and 334 host RNAs that interact with the WT and Δ382 SARS-CoV-2 genomes, respectively (Fig. 5a, b, Supplementary Fig. 13a–c, Supplementary Data 8,9). The host RNA-virus genome interactions are preferentially enriched in the coding regions along host mRNAs (Fig. 5c). STRING analysis of the top 25% of SARS-CoV-2 interactors

showed that they are enriched for proteins that physically interact with each other (PPI: $p < 10^{-16}$)[37], including genes that are involved in the mitochondria, ER, GTP hydrolysis, and translation processes (Fig. 5d). GO term enrichment of interacting RNAs showed similar enrichments, confirming the importance of SARS-CoV-2 interactors in mitochondrial and ER function (Fig. 5e)[38,39].

While SARS-CoV-2 RNAs bind to more than 300 RNAs inside cells, we observed that the top 10 (2.6%) of the strongest interactors contributed to 17.5% and 24.1% of all WT and Δ382 binding events, indicating that the virus binds to them particularly strongly (Fig. 5b). These strong interactors include mitochondrial RNAs such as the mRNA of COX1, which is a mitochondrially encoded cytochrome-c oxidase, mitochondrial rRNA and tRNA, and SNORD27, a snoRNA responsible for 18 S ribosomal RNA methylation (Supplementary Fig. 13b). Using the program RNAcofold, we observed strong pair-wise interactions between virus and mitochondria RNAs in SPLASH identified binding sites (Supplementary Fig. 13d). While generally, SARS-CoV-2 interacts stronger with more abundant host RNAs, we observed significantly more interactions between the virus and host mitochondrial and snoRNAs than expected from abundance alone (Supplementary Fig. 13e). A previous study using RNA-GPS had shown that part of SARS-CoV-2 RNAs localized to the mitochondria and the nucleolus[40]. SARS-CoV-2 infection also results in mitochondrial dysregulation[23,41]. Further experiments are needed to test whether the direct pairing between SARS-CoV-2 and mitochondrial RNAs contributes to mitochondrial dysregulation.

The SARS-CoV-2 infection has been found to have an impact on almost every aspect of the host transcriptome to control virus and host gene regulation[42]. We observed a general decrease of RNA abundance of SARS-CoV-2 interactors upon virus infection. Interestingly, however, an opposite trend was observed for the strong interactors that were selectively stabilized and their abundance is increased upon SARS-CoV-2 infection (Fig. 5f, g, Supplementary Fig. 13f, g). qRT-PCR analysis of key interactors such as COX1 mRNA and MT-rRNA showed that these RNAs are indeed stabilized upon virus infection, confirming our RNA sequencing results (Supplementary Fig. 13h–k). Mining of published SARS-CoV-2 proteomics data revealed that proteins encoded by SARS-CoV-2 interactors were also preferentially translated and/or stabilized at the protein level as compared to proteins produced by non-SARS-CoV-2 interactors (Fig. 5h)[43]. Thus, interaction with SARS-CoV-2 RNAs may confer a stabilizing effect on their overall gene regulation.

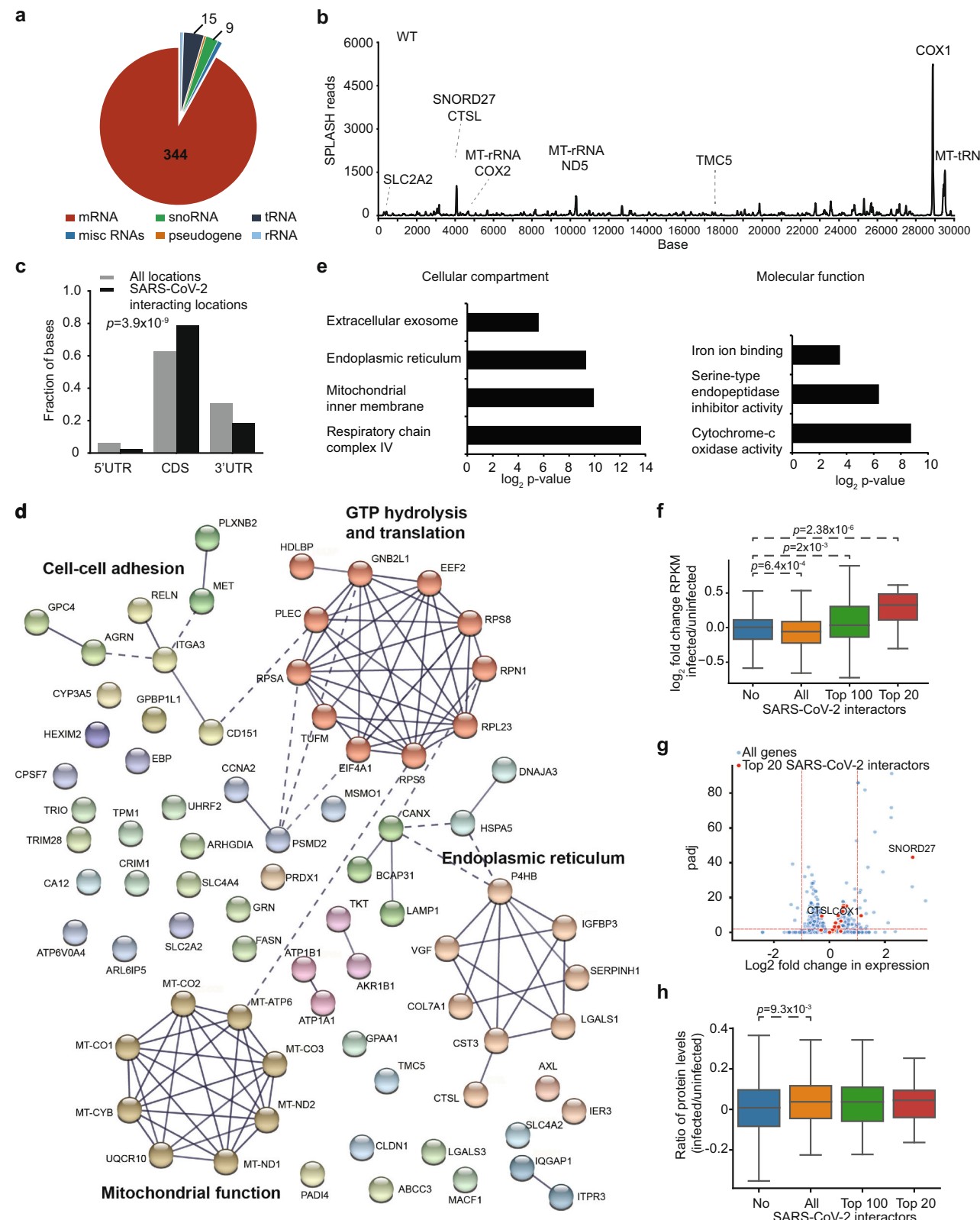

**SARS-CoV-2 RNA binds to SNORD27 and is 2'-O-methylated.** SNORD27 is one of the strongest host interaction partners for SARS-CoV-2 RNA (Fig. 6a) and is traditionally known to guide 2'-O-methylation of 18 S ribosomal RNA[44]. snoRNAs can bind and methylate cellular RNAs[45], and methylation enzymes including fibrillarin (FBL), rRNA methyltransferase 2 and 3 (MRM2 and MRM3) have been found to be physically associated with SARS-CoV-2 genome[23]. We tested whether SARS-CoV-2 RNA could be 2'-O-methylated and whether host RNAs' methylation levels are changed upon virus infection. We

**Fig. 5 SARS-CoV-2 interacts with hundreds of host RNAs in vivo. a** Pie-chart showing the number of host RNAs from different RNA classes that interact with WT SARS-CoV-2. **b** Line plot showing the number of SPLASH reads along the WT SARS-CoV-2 genome. The names of host RNAs that bind strongly to the virus at a particular location is labelled above the interaction peak. **c** Bar-chart showing the fraction of host interacting regions that fall in 5′ UTR, CDS, and 3′ UTR (black), as compared to what is expected from random (grey). Host interacting regions are enriched in CDS and depleted in 3′ UTRs. Significance was assessed using a one-way Chi-Square test without adjustments. **d** STRING analysis of the top 25% SARS-CoV-2 host interactors[37]. The networks were built based on high confidence (0.7) evidence from protein-protein interaction sources of experiments, databases, and text-mining where the line thickness indicates the strength of data support. Functional clusters in PPI networks were determined using the Markov Clustering algorithm (MCL). The PPI enrichment $p$-value $< 10^{-16}$. **e** GO term enrichment of the top 25% SARS-CoV-2 interactors using David functional annotation analysis. SARS-CoV-2 interactors are enriched for transcripts that reside in the mitochondria, ER, and exosome, and are enriched for molecular functions for iron-binding, endopeptidase inhibitor activity, and cytochrome-c oxidation activities. The $p$-value was calculated using a hypergeometric test. **f** Boxplots showing the distribution of log2 fold change in gene expression upon SARS-CoV-2 infection in non-interacting genes, in 374 RNAs that interact with SARS-CoV-2 (All), in the top 100 interactors and top 20 interactors ranking by the chimeric read counts. SARS-CoV-2 interactors show a decrease in gene expression upon virus infection. However, the top interactors show an increase in gene expression upon virus infection, indicating that they are selectively stabilized. The expression data were calculated from the non-chimeric reads from SPLASH and quantified using DESeq2[58]. The $p$-value was calculated using the two-tailed Wilcoxon Rank Sum test. The box represents the 25–75th percentiles, and the median is indicated. The whiskers show the minimum and maximum values. **g** Volcano plot showing the distribution of host RNA gene expression upon SARS-CoV-2 infection. The top 20 interactors are highlighted in red and show a general stabilization in gene expression upon virus infection. The $p$-value was calculated using the two-tailed Wilcoxon Rank Sum test. **h** Boxplots showing the distribution of protein ratio after virus infection in all genes, in 374 RNAs that interact with SARS-CoV-2, in the top 100 interactors and top 20 interactors. The $p$-value was calculated using the two-tailed Wilcoxon Rank Sum test. The box represents the 25–75th percentiles, and the median is indicated. The whiskers show the minimum and maximum values. Source data are provided as a Source Data file.

performed 2 biological replicates of Nm-seq on total RNA from HeLa cells, as well as from uninfected and SARS-CoV-2 infected Vero-E6 cells (Supplementary Fig. 14a, Supplementary Data 10,11, Methods)[46]. Biological replicates of Nm-seq from both cell types show that they are well correlated, suggesting that Nm-seq data is reproducible (Supplementary Fig. 14b,c). Nm-seq analysis on human 18 S rRNA accurately identified 36 out of 42 known 2′-O-methylation sites and had a high AUC-ROC curve of 0.96, suggesting that we are able to detect existing 2′-O-methylation sites accurately and sensitively (Supplementary Fig. 14d,e).

Using Nm-seq, we identified a total of 130 2′-O-methylation sites in the SARS-CoV-2 genome (Fig. 6b), and 4931 sites in 4142 transcripts in the Vero-E6 transcriptome (Supplementary Fig. 14f,g). We observed that a 2′-O-methylated host mRNA contains approximately 1.1 modifications per transcript in the Vero-E6 transcriptome, similar to methylated RNAs in HeLa cells[46]. The majority of these host modification sites (60%) were located in the coding regions and were enriched for codons encoding charged amino acids (Supplementary Fig. 15a), as previously described[46]. In comparison, the SARS-CoV-2 genome is 19-fold more modified than host mRNAs after normalizing for transcript length (Fig. 6c). The 2′-O-methylations are enriched in the 5′ and 3′ UTRs of SARS-CoV-2 (Fig. 6d), depleted in position 2 of codons (Supplementary Fig. 15b), and are enriched for U and depleted for G bases along the genome (Fig. 6e). 2′-O-methylation sites on SARS-CoV-2 are also associated with high SPLASH reads, indicating that they are located near positions with abundant intramolecular pair-wise interactions (Fig. 6f). As bases that are 2′-O-methylated cannot be modified by NAI, we tested whether 2′-O-methylated bases have lower SHAPE-MaP reactivity. We did not observe a decrease in SHAPE-MaP reactivity in 2′-O-methylated bases as compared to non-methylated bases (Supplementary Fig. 15c), suggesting that only a fraction of the SARS-CoV-2 genome at the base is modified.

As the modification of the SARS-CoV-2 genome might sequester corresponding RNA modification enzymes away from the host transcriptome, we calculated the changes in modification rates in the host RNAs in the presence and absence of SARS-CoV-2 infection. We observed a decrease in host RNA 2′-O-methylation frequency upon virus infection, supporting our hypothesis that they become less methylated (Fig. 6g). In addition, we also observed that host RNAs that interact strongly with the SARS-CoV-2 genome

show greater 2′-O-methylation changes, and show large losses and gains in modification sites within the RNAs (Fig. 6h, i, Supplementary Fig. 15d, e). We hypothesized that RNAs interacting with the SARS-CoV-2 genome could be methylated near their interacting regions, presumably due to proximity to SNORD27, while methylation sites that are located far away could be lost. To determine whether 2′-O-methylation sites on SARS-CoV-2 genome interacting RNAs are closer to the locations of virus-RNA interactions regions, we calculated the closest distance between virus-host interaction to host 2′-O-methylation site. We observed that sites that had 2′-O-methylation were indeed closer to virus-host RNA interactions sites (Fig. 6j), supporting the hypothesis that proximity to the SARS-CoV-2 genome might allow interacting RNAs to be methylated together within a hub.

2′-O-methylation has been shown to stabilize RNAs inside cells[45]. Therefore, we hypothesized that the loss of 2′-O-methylation on host RNA upon virus infection may affect the stability of the host RNAs. Indeed, we observed that the abundance of host RNAs that show a decrease in methylation sites was significantly decreased in infected cells. In contrast, the abundance of host RNAs that show an increase in methylation sites was increased (Fig. 6k). Thus, together with the production of Nsp1 from SARS-CoV-2 to cleave cellular RNAs, the binding of SARS-CoV-2 RNAs to SNORD27 could serve as part of a multi-prong mechanism to decrease cellular RNAs and maximize virus replication (Fig. 6l)[47].

## Discussion

Studying the molecular basis of virus pathogenicity enables us to understand how this can be counteracted and how to inhibit and target the replicating virus. By probing the local and pair-wise RNA interactions of the SARS-CoV-2 genome and sgRNAs using high throughput structure probing technologies on both the Illumina and Nanopore sequencing platforms, we identified potentially functional structure elements within the genome and demonstrated that these RNA structures are associated with ribosome pausing. While the structures along SARS-CoV-2 RNA have also been probed using other short-read high throughput strategies including icSHAPE and DMS-MapSeq[29], these strategies have limitations in their ability to decipher sgRNA-specific structures due to extensive sequence similarity between the different sgRNAs. Using long-read sequencing, we identified both

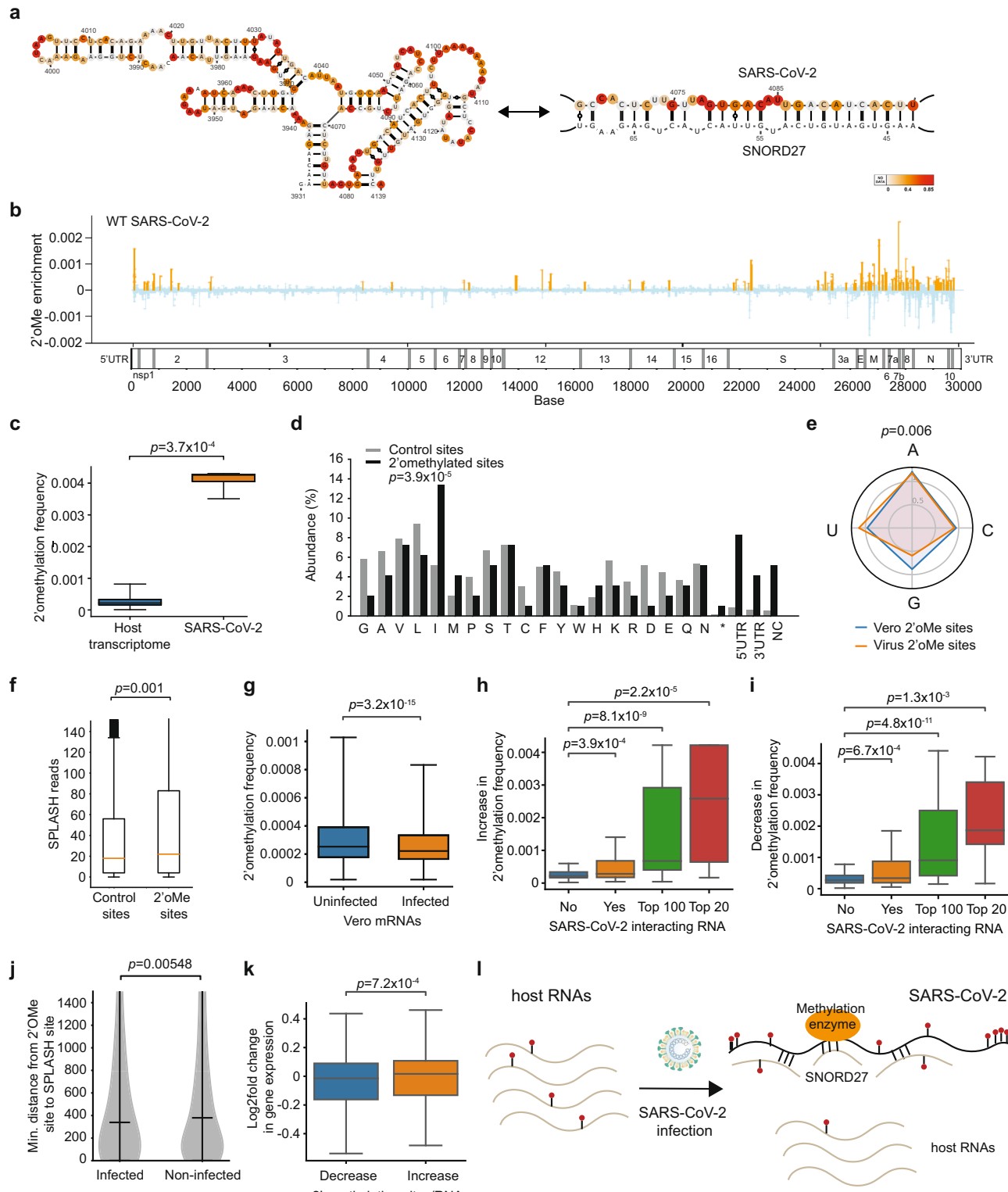

sgRNA-specific structures, as well as structure differences between WT and Δ382 genomes, that could serve as a basis for understanding sgRNA-specific functions in the future.

While existing literature has mostly focused on understanding SARS-CoV-2: host protein interactions, here we describe that the virus genomes bind directly to hundreds of host RNAs inside cells using SPLASH. In addition to a previous report that showed that SARS-CoV-2 binds to snRNAs that are involved in splicing[36], we identified diverse, functionally related, host mRNA-virus

interactions and found that SARS-CoV-2 binds particularly strongly to mitochondrial RNAs and snoRNAs. Our results are consistent with previous predictions of SARS-CoV-2 localization in the mitochondria and nucleolus[40], and the observation that the mitochondria is dysregulated upon SARS-CoV-2 infection[23]. In addition, previous studies have also shown that SNORD27 and mitochondrial RNAs are enriched on the SARS-CoV-2 genome when the genome is isolated using formaldehyde crosslinking and sequencing[23]. However, it remained unclear if these RNAs bind

**Fig. 6 2'-O-methylation of SARS-CoV-2 sequesters methylation away from host RNAs. a** Structure model of SARS-CoV-2 RNA before and after SNORD27 binding. The model of SARS-CoV-2 before SNORD27 binding is generated using the RNA structure program[54] and incorporating SHAPE-MaP reactivity as constraints. The model of SARS-CoV-2:SNORD27 pairing is generated using RNAcofold[35]. SHAPE-MaP reactivities are mapped onto the structure models. **b** The distribution of 130 2'-O-methylation sites along WT SARS-CoV-2 genome. The y-axis is the 2'-O-methylation enrichment measured by the RT-stop fraction against the coverage at each nucleotide. The orange bars indicate enriched sites above control. **c** Boxplots showing the distribution of 2'-O-methylation frequency along with the host RNAs and on SARS-CoV-2 RNA from $n = 2$ biological replicates (total 4 technical replicates). P-value was calculated using the two-tailed Wilcoxon Rank Sum test. **d** Bar-plot showing the distribution of 2'-O-methylation sites (black), and control sites (grey), in the different amino acids, 5' UTR, 3' UTR, and non-coding regions (NC) along the SARS-CoV-2 genome. P-value was calculated using the chi-squared test. **e** Distribution of 2'-O-methylation sites on SARS-CoV-2 genome (orange) and Vero-E6 transcriptome (blue) at A, C, U, G bases. The proportion of each nucleotide was normalized by its prevalence in the host transcriptome and SARS-CoV-2 genome, respectively. 2'-O-methylation sites on SARS-CoV-2 are enriched in Us and depleted in Gs. P-value was calculated using the chi-squared test. **f** Boxplot showing the distribution of SPLASH chimeric reads at all sites ($n = 29,847$) versus 2'-O-methylation sites ($n = 485$) along the SARS-CoV-2 genome. P-value was calculated using the two-tailed Wilcoxon Rank Sum test. **g** Boxplot showing the distribution of 2'-O-methylation frequency along Vero-E6 mRNAs in uninfected and SARS-CoV-2 infected cells. P-value was calculated using the two-tailed Wilcoxon Rank Sum test. **h, i** Boxplot showing the distribution of increase (**h**) or decrease (**i**) in methylation frequency in non-interacting RNAs, in all interacting RNAs, in top 100 interacting RNAs, and in top 20 interacting RNAs. P-value was calculated using the two-tailed Wilcoxon Rank Sum test. **j** Violin plot showing the distribution of the minimum distance between host 2'-O-methylation site and the location of its interaction with SARS-CoV-2. P-value is calculated using the Wilcoxon Rank Sum test. **k** Boxplot showing the distribution of changes in gene expression in Vero-E6 mRNAs that either lost (decrease) or gained (increase) 2'-O-methylation sites. P-value is calculated using the Wilcoxon Rank Sum test. In (**c**, **f–i**, and **k**), the box represents the 25–75th percentiles, and the median is indicated. The whiskers show the minimum and maximum values. **l** Model of our hypothesis. SARS-CoV-2 binds to SNORD27 to sequester methylation enzymes to itself and away from host mRNA, enhancing host RNA decay. Source data are provided as a Source Data file.

to the virus genome directly or indirectly through protein interactions. Our studies show that SARS-CoV-2 RNA pairs directly with mitochondrial RNAs and snoRNAs at specific locations on host RNAs and SARS-CoV-2 genomes. As snoRNAs recruit 2'-O-methylation modifications on their target RNAs, we additionally observed that the SARS-CoV-2 genome is extensively 2'-O-methylated inside cells. Interestingly, the 2'-O-methylation sites do not coincide with the location of SNORD27 binding. One hypothesis is that SNORD27 could recruit methylation enzymes and act as a hub to methylate spatially proximal regions that are far away in linear space. While more studies need to be performed to fully comprehend the mechanism of 2'-O-methylation on the SARS-CoV-2 genome, this study deepens our understanding of SARS-CoV-2 biology and supports the observation that SARS-CoV-2 interacts with methylation enzymes, including FBL, inside cells.

2'-O-methylation of RNA plays important roles inside cells and can contribute to RNA stabilization as well as key functions in innate immunity[48]. A recent study revealed that HIV-1 hijacks cellular proteins to 2'-O-methylate its genome to escape from host innate immune sensing by modulating the expression of type-1 interferons[49]. Further experiments are needed to determine if 2'-O-methylation of SARS-CoV-2 RNA could also allow it to escape host immunity. Another hypothesis for the binding of SARS-CoV-2 to SNORD27 is that the virus sequesters SNORD27 and methylation complexes towards itself and away from the rest of the Vero-E6 cell transcriptome. We observed that 2'-O-methylation sites on SARS-CoV-2 are enriched for paired interactions, in agreement with previous literature that 2'-O-methylation in HIV-1 stabilizes alternative pairing confirmation of the transactivation response element[50]. Importantly, we observed that 2'-O-methylation levels in host RNAs decrease after SARS-CoV-2 infection, supporting our hypothesis that the binding of SNORD27 to SARS-CoV-2 could direct methylation enzymes to SARS-CoV-2, and away from host RNAs. In addition to the function of SARS-CoV-2 Nsp1 to degrade host RNAs[47], this could serve as part of a multi-prong strategy for the virus to degrade host RNA for its own benefit. Further experiments would be needed to definitively prove this hypothesis.

In summary, our study identifies new potentially functional structures along the SARS-CoV-2 genome, new host factors, and alternations of host gene regulation upon SARS-CoV-2 infection,

providing a critical new understanding of the SARS-CoV-2 infection process.

## Methods

**Cells and viruses.** African green monkey kidney, clone E6 (Vero-E6) cells (ATCC# CRL-1586) were maintained in Dulbecco's Modified Eagle Medium (DMEM) supplemented with 5% fetal bovine serum (FBS). HeLa cells (obtained from neighboring labs in GIS) were grown in DMEM high-glucose media (Thermo Fisher Scientific), supplemented with 10% FBS, 1% Pen-Step. SARS-CoV-2 wild-type (hCoV-19/Singapore/2/2020, GISAID accession ID: EPI_ISL_407987) and Δ382 mutant (hCoV-19/Singapore/12/2020, GISAID accession ID: EPI_ISL_414378) were isolated from COVID-19 patients in Singapore, as reported previously[7].

**SHAPE-MaP structure probing of SARS-CoV-2 virus in Vero-E6 cells.** Vero-E6 cells were infected with SARS-CoV-2 viruses (WT and Δ382) at a multiplicity of infection (MOI) = 0.01 for 1 h at 37 °C. Following 1 h infection, virus inoculum was removed and replaced with DMEM-5% FBS. Flasks were incubated for 48 h at 37 °C, 5% $CO_2$.

At 48 hpi, cells were washed once with PBS and trypsin was added to detach the cells from the flask. The cells were collected and centrifuged at $300 \times g$ for 5 min. The pellet was resuspended in PBS and the cells were then separated into three reactions: (1) added 1:20 volume of 1 M NAI (03-310, Merck, 25 μl of NAI in 500 μl of infected cells) and incubated for 15 min at 37 °C for structure probing; (2) added 1:20 volume of dimethyl sulfoxide (DMSO) and incubated for 15 min at 37 °C, as negative control; and (3) set aside a third portion of the infected cells without any treatment, as the denaturing control in the downstream library preparation process. The total RNA was extracted from the cells using E.Z.N.A. Total RNA Kit (Omega bio-tek) according to the manufacturer's instructions. We then performed library preparation following the SHAPE-MaP protocol to generate cDNA libraries compatible for Illumina sequencing[18].

**Interactome mapping of SARS-CoV-2 virus in Vero-E6 cells.** Vero-E6 cells were infected with SARS-CoV-2 viruses (WT and Δ382) at a multiplicity of infection (MOI) = 0.01 for 48 h. The cells were washed once with PBS and trypsin was added to detach the cells from the flask. The cells were collected and centrifuged at $300 \times g$ for 5 min. The pellet was resuspended in PBS and the cells were then incubated with 200 μM biotinylated psoralen and 0.01% digitonin in PBS for 10 min at 37 °C. The cells were spread onto a 10 cm dish and irradiated at 365 nm of UV on ice for 20 min. The cells were collected, and the total RNA was then extracted using E.Z.N.A. Total RNA Kit (Omega bio-tek) according to the manufacturer's instructions. We performed SPLASH libraries similarly to the published protocol[20,51]. The qRT-PCR primer sequences are listed in Supplementary Table 1.

**Direct RNA sequencing using Nanopore.** Unmodified and NAI-treated total RNA from WT and Δ382 SARS-CoV-2 infected Vero-E6 cells were sequenced using Nanopore direct RNA sequencing 002 kit. The samples are sequenced and aligned according to the method used by Kim et al. [9]. We used EPI_ISL_407987 and EPI_ISL_414378 as the reference for WT and Δ382 strain, respectively.

**Nm-Seq library construction**. To map RNA nucleotides with 2'-O-methyl modification, Nm-Seq[51] was applied to the total RNA of HeLa, Vero-E6, and Vero-E6 infected with SARS-CoV-2. In brief, eight rounds of oxidation-elimination-dephosphorylation (OED) were performed to iteratively eliminate non-modification nucleotides from the 3' ends of fragmented RNAs, the 2'-O-methylated nucleotides resist the OED and make the 3' end of reads enriched at the 2'-O-methylation sites. Two biological replicates of SARS-CoV-2 infected Vero-E6 (total 4 replicates) and uninfected Vero-E6 (total 2 replicates) were used to generate Nm-Seq libraries. Ten μg of each sample was used following NmSeq protocol and NEBNext Small RNA library kit with revised customized adaptors[46,52] (IDT). Customized 3' SR adaptor: 5'-AppNN NNN ATC ACG AGA TCG GAA GAG CAC ACG TCT-3'. Customized 5' SR adaptor: 5'-GUU CAG AGU UCU ACA GUC CGA CGA UC NNNNN-3'. For the input control library without OED, 1 μg of RNA were used. Libraries were multiplexed and subjected to high-throughput sequencing using Illumina Next-Seq Hi.

**Data analysis**. Processing, analysis, and visualization of data were performed using python-3.6.8, R-3.4.1, and the associated modules numpy-1.17.3, scipy-1.3.1, matplotlib-3.1.2, and R-scape 1.4.0. Calculation of statistical parameters of data sets (including means, medians, percentiles, and standard deviations) was performed using numpy. Statistical tests (t-tests, chi-square tests, and Wilcoxon Rank Sum tests) employed the appropriate functions in the scipy.stats module. Visualization was performed using matplotlib for contour plots, bar charts, pie charts, box plots, and violin plots unless otherwise specified.

**Analysis of SHAPE-MaP experiments**. Sequencing reads obtained from two replicates of SHAPE experiments were aligned with the respective sequences for the strains (WT: EPI_ISL_407987, Δ382: EPI_ISL_414378) and SHAPE values for each position calculated using 'Shapemapper-2.15' of Weeks et al. [53] using Bowtie-2.4.2 for reading alignment. Read depths obtained in the sequencing experiments allowed for the conclusive determination of SHAPE reactivities at approximately 80% of positions. A reference alignment of WT and Δ382 sequences were obtained using 'mafft-7.453' with the L-INS-I strategy. Subsequently, the local correlation of SHAPE reactivity between replicates and between WT and Δ382 was calculated using Pearson correlation. As Pearson correlation between replicates was > 0.9, replicates were pooled for subsequent analysis.

**Modelling of global RNA structure**. Using the above SHAPE reactivities and reference sequences, separate global RNA structure models for WT and Δ382 using 'Superfold' with a maximum base-pairing distance of 600 nt, and default SHAPE slope (1.8) and intercept (−0.6) parameters[54]. Regions of interest such as the frameshift element were modelled separately in a local context to search for pseudoknot structures using the 'RNAstructure' tools 'partition-smp' and 'Prob-Knot-smp'. Additionally, we used ScanFold to assess likely local fold stability. We used a step size of 10, a window size of 120, and 50 randomizations. Regions were ranked by Z-score and the lowest 20% Z-scores, corresponding to the highest stabilizations, were identified (Fig. 2a).

**Modelling of individual sgRNA structure from Nanopore data**. The distributions of mutation rates obtained from Nanopore RNA sequencing were compared with the distribution of SHAPE reactivity values from short-read-based experiments described above. We found that a scale factor of 100 brings the mutation rate distribution from the Nanopore experiment in line with the distribution observed for conventional SHAPE experiments. Hence, we applied this scaling factor and then employed these as SHAPE data in the same 'Superfold' protocol as for the full-length models described above[54].

**PORE-cupine analysis of direct RNA sequencing data**. Filtering for full-length sgRNA: To separate full-length aligned reads into their sub-genomic transcripts, we used two filtering conditions for all sgRNAs except for ORF6. One, the aligned reads need to contain the leader sequence, and two, the aligned positions after the leader sequence must fall within ± 100 of the annotated sgRNA sequences. For ORF6, we had to extend the second filter to −300 of the annotated sgRNA sequence.

We calculated the reactivity for each subgenomic transcript by using PORE-cupine 1.0, with two adjustments to the analysis. (1) The length filter was removed, as only full-length transcripts were used for the analysis. (2) To reduce the amount of computing resources required, 20,000 strands from each subgenomic transcripts in the unmodified libraries were randomly selected and used for the generation of models.

To determine the differences between reactivity, the Wilcoxon Rank Sum test was applied to a 101 bases window with a step size of 25 nucleotides. Reactivity differences were compared across shared sequences between the different subgenomic transcripts within each strain (WT or Δ382), and across the two different strains. For the comparison between WT and Δ382 genome, the region in the WT strain that was not present in the Δ382 strain was masked. The p-values are corrected with Hommel's method. In addition to using the statistical test to determine the differences, we added the second criteria of Pearson correlation < 0.7.

**Analysis of SPLASH experiments**. Chimeric reads were divided into host-host, host-virus and virus-virus interactions for WT and Δ382 genomes. Virus-virus interactions were normalized to total virus-virus interactions and are shown in Fig. 3a (WT: blue, Δ382: red). Virus-host interactions were equally normalized and the main locations for host interactions are shown in Fig. 5b. SPLASH hybrid structure models are generated using RNAcofold in the Vienna RNA package.

**Protein-protein interaction network analysis by STRING**. The top 25% of host RNA interactors with SARS-CoV-2 were used as input for STRING analysis[37], a search tool for retrieval of interacting genes to acquire protein-protein interaction (PPI) networks. The networks were built based on high confidence (0.7) evidence from protein-protein interaction sources of experiments, databases, and text-mining where the line thickness indicates the strength of data support. Functional clusters in PPI networks were determined using the Markov Clustering algorithm (MCL). The PPI enrichment p-value < 1.0e−16.

**Nm-Seq bioinformatics analysis**. We referred to the pipeline from the published protocol with some modifications[52]. The adaptors on raw reads were trimmed using Cutadapt and the reads without 3' adaptors were discarded[55]. The PCR duplication reads were filtered out by the pentamers at the 5' and 3' end of the reads as barcodes using a custom script. The pentamers were then removed and reads that are shorter than 15 nt were discarded. The reads that passed all these filters were mapped to the reference which combined the longest transcriptome of *Chlorocebus sabaeus* (ensembl ChlSab1.1.101) and SARS-CoV-2 sequence (WT_EPI_ISL_407987) by bwa-aln[56]. The multiple mapped reads and reads with soft and hard clipped alignments were discarded. The depth of each position on the transcriptome using 3' end of reads was calculated in both input control and OED enriched libraries ($D_{input}$, $D_{OED}$). The depths were normalized by the read counts of each transcript. The significantly enriched sites over the other regions on the same transcript were detected by $\Delta D = D_{OED} - D_{input}$ using Z-test (FDR < 0.05, $D_{OED} >= 10$, $D_{OED}/D_{input} > 1.5$).

As the SARS-CoV-2 transcripts comprise 16–30% of the total reads, we subsampled the two uninfected libraries into 4 libraries containing the same reads on the host transcriptome with the 4 infected replicates respectively, to compare the Nm sites of host transcriptome fairly. Bases that show 2'-O-methylation enrichment in 3 out of 4 replicates were recognized as the 2'-O-methylation sites.

**Reporting summary**. Further information on research design is available in the Nature Research Reporting Summary linked to this article.

## Data availability
The data sets generated during and/or analysed during the current study are available in the GEO repository, GSE165005. The ribosome profiling data is obtained from a published study[34] (https://doi.org/10.1038/s41586-020-2739-1). The proteomics data is obtained from published study[43] (https://doi.org/10.1038/s41586-020-2332-7). The authors declare that all other data supporting the findings of this study are available within the article and its Supplementary Information files. Source data are provided with this paper.

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

## Acknowledgements

We thank members of the Wan lab for helpful discussions. Y. W. is supported by funding from A*STAR, NRF, EMBO Young Investigatorship and CIFAR global fellow scholarship. This project was funded in part by National Medical Research Council, Singapore COVID19RF2-0001 to D.E.A. and L.F.W.

## Author contributions

Y. W. conceived the project. Y.W., R.G.H., L.F.W., A.L., and A.M. designed the experiments and analysis. S.L.Y., D.E.A., A.A., S.Y.L., X.N.L., K.Y.T., T.C., and Y.S. performed the experiments. R.G.H., L.D., Y.Z., A.A. and T.Z. performed the computational analysis. Y.W. organized and wrote the paper with R.G.H. and all authors.

## Competing interests

The authors declare no competing interests.
