## [Peer Review File · Nature Communications]

Reviewer comments, first round –

Reviewer #1 (Remarks to the Author):

This manuscript from Yang, DeFalco, Anderson, et al. is a magisterial study of SARS-CoV-2 that uses multiple orthogonal approaches to dissect specific viral/host interactions and RNA structure. Their structural analysis adds to the field despite other labs already having published DMS/SHAPE MaPseq experiments. They include psoralen and nanopore sequencing to distinguish long-range interactions and data from individual sgRNAs. Their discovery of interactions with host factors and the corresponding changes in host cell biology are interesting, particularly the increase in 2'OH methylation of SARS-CoV-2 genome and a corresponding decrease in host RNA 2'OH methylation. They provide a good explanation for why this may be occurring. Overall, the manuscript is very well written, and I think this will be a great addition to Nat. Comm. Just a few comments follow below.

Major Comments:

Due to the high level of 2'OH methylation of SARS-CoV-2 RNA, do the authors think this could be affecting SHAPE probing experiments in any way (as the site is no longer likely to be reactive with NAI)? Would DMS probing provide additional information here? An inclusion of this point in the discussion would be nice.

Line 233-235: I would expect the overall correlation of structure between the two models to be larger, given it is such a small section of the genome (~1.5%) that is changed. Could more discussion be offered here as to the change in overall structure?

It's great that the authors applied ScanFold as part of their analytical pipeline. It would be very helpful, however, to include a better description of what was actually done (window size, step size and shuffling protocol) to predict low energy regions. A description of what is meant by low energy region (low z-score?) is also important. The authors could consider referencing two previous applications of ScanFold to SARS-CoV-2:

bioRxiv. 2020 Apr 18;2020.04.17.045161. doi: 10.1101/2020.04.17.045161.

Mol Cell. 2021 Feb 4;81(3):584-598.e5. doi: 10.1016/j.molcel.2020.12.041.

Minor comments:

Line 37: "SARS-CoV-2 binds", should say "SARS-CoV-2 genome" so people don't confuse it with the fully formed virus.

Lines 131 and 136: " local 'Scan-Fold' energies" this can just be ScanFold

Lines 175-179 and 287-288 could use a bit of editing.

Line 499: there is a "," after the 1. In the next line they use a "." after the 2. Make these consistent and perhaps use ")") to better distinguish it?

Finally, they refer to the program RNAstructure as "RNA structure" several times, just remove the space.

-Walter Moss

Reviewer #2 (Remarks to the Author):

In this manuscript, Yang et al applied three different structure-mapping methods to interrogate the RNA structure of the SARS-CoV-2 genome and the previously characterised 382-nucleotides deletion version of the genome, inside Vero cells (monkey origin). The authors identified previously characterised structural elements, report on long-range interactions along the genome and discover interactions between the virus genome and host RNA. Using a long-read sequencing platform, the authors identified some differences between the structure of different subgenomic RNAs. The authors identified binding between the virus genome and a host SnoRNA involved in 2'-O-methylation. Using a technique to map 2'-O-methylation sites on RNA, the authors identified several methylation hotspots along the virus genome.

Overall, the findings are important and well controlled. Some of the structural models are in agreement with previous publications, while others (for example, direct interactions with snoRNAs) are novel and very interesting. I therefore recommend publication in Nature Communications pending minor revisions, below.

1. Several different structural models for the SARS-CoV-2 frameshifting element have been recently suggested, including [doi.org/10.1016/j.molcel.2020.12.041], [doi.org/10.1101/2020.06.29.178343], and [doi.org/10.1074/jbc.AC120.013449]. Wherever possible, the authors should align their structural data to each of the current models and show whether one of the previously suggested conformations is more supported by their data. There are minor differences between the frameshifting element structures shown in figure 1 and supplementary figure 2. The authors should explain the reason for these differences.
2. The manuscript would benefit from adding some background on the function of the frameshifting-element.
3. The authors used their previously developed SPLASH method to analyse RNA-base pairing along the virus genome. It would be important to detail the depth of the data (how many chimeric reads have been identified overall) and the type of control used for these measurements.
4. The authors compare the structure of the wild type and the deletion version of the virus. However the correlation between the SHAPE reactivity of these two genomes (supplementary figure 8a) seems greater than the level of reproducibility between replicates of the same genome (supp. Figure 1C). The identified differences would therefore benefit from appropriate statistical analysis.
5. The comparison between the structure of the virus genome and the different subgenomes would also benefit from statistical analysis. The authors should mention that some structural differences between the SARS-CoV-2 genome and subgenomes were previously identified using a pulldown strategy to separate genomes from subgenomes [doi.org/10.1016/j.molcel.2020.11.004].
6. The interaction between SNORD27 and the viral genome does not seem to coincide with the identified 2'-O-methylation sites. The authors should refer to that in their discussion.

Reviewer #3 (Remarks to the Author):

In the manuscript "Comprehensive mapping of SARS-CoV-2 interactions in vivo reveals functional virus-host interactions", the authors use SHAPE mutational mapping, PORE-cupine, SPLASH and Nm-seq to identify novel divergent gRNA structures, compare structure and folding of WT and Δ 382 virus genomes, identify direct virus and host-genome interactions, and explore potential effects of the virus on gene regulation of the host. There are a few instances where the authors do not fully explain their reasoning for or the implications of a finding, and where they overstate, but overall this seems like a novel and significant contribution to the understanding of SARS-CoV-2

infection and interaction with the host, and provides a direction for future studies of viral mechanism.

Comments:

1. Fig 2b: what do these highly structured elements tell us? The use for ssRNA regions that could be targeted is clear, but what can be predicted about the function of the novel structured elements that are displayed here? I'd add that the way the structures for the RNA are plotted makes it difficult to immediately derive meaning - I'd suggest showing on or two specifically interesting regions, along with a clearer annotation of where the siRNA target sites are and what gene they are from.
2. Fig 3a: Is there a way to quantify % of interactions that are the same in WT versus delta382 genomes? It appears robust, but statistics on the correlation of interactions would be helpful. Perhaps an upset plot or venn diagram with a filtered strength of interaction?
3. Fig 3d: ribosome pause sites do not have significantly more SPLASH reads; also what occupancy is included in the "all sites"; could they provide a list? Why are the reads so low compared to the other interaction length data; is it because of the ribosome profiling dataset? Why not compare the pause sites to non-pause sites versus all sites?
4. There are regions where the SHAPE-MaP and PORE-cupine data do not correlate well (Supp 5d) - any discussion of this? The R^2 is 0.49, which I wouldn't necessarily consider to be a good correlation.
5. Figure 4 is novel in that the authors identify different structure models for sgRNAs, but biological context would be helpful; how does this compare to other viruses - even if predicted, what does this mean for viral-genome interactions that are shown here, why did they want to determine the structures for these sgRNAs?
6. Can the authors conjecture why the top 10 strongest interactors contributed to slightly more delta382 binding events than WT binding events? Are additional binding events lost in the delta382 strain that would cause the others to be higher contributors?
7. I'm curious about the SPLASH interaction between the viral and host RNAs - specifically I'm interested in how normalization/controls were done to account for the proposed strong interaction with mitochondrial RNA. How does this compare to other splash data? Is it normalized to account for the expression level of mito RNAs? Do other exogenously expressed RNAs or other viruses show a similar type of interaction?
8. In Fig6, can the authors conjecture why 2'-O-methylation might also be enriched at I codons in addition to the 5' and 3' UTRs?
9. The authors say that the distance from the 2'-O-methylation site are closer to virus-host interaction sites, but by eye it looks pretty similar between uninfected and infected cells, and the p-value is right on the edge of significance....maybe a more interesting way to look at this is to think about the minimum distance overall between the sites of 2'-O-methylation and potential interaction sites, whether or not the cells are infected.
10. In 6k, though there is an decrease/increase in abundance of RNAs, but static information cannot give information about decay, so they just have to be careful about their claims; perhaps there could be a feedback loop and decrease in production as well.
11. Finally, given that the SHAPE reagents target the 2'-hydroxyl site as well, I'm curious if the 2'-O' methylation is detectable from the shape-map or PORE-cupine results? Is it blocked or otherwise detectable through a shift in reactivity?

We thank the reviewer for their positive and helpful comments, which has made our manuscript clearer and better in quality. Below, we address the reviewers' comments point by point.

REVIEWER COMMENTS

Reviewer #1 (Remarks to the Author):

This manuscript from Yang, DeFalco, Anderson, et al. is a magisterial study of SARS-CoV-2 that uses multiple orthogonal approaches to dissect specific viral/host interactions and RNA structure. Their structural analysis adds to the field despite other labs already having published DMS/SHAPE MaPseq experiments. They include psoralen and nanopore sequencing to distinguish long-range interactions and data from individual sgRNAs. Their discovery of interactions with host factors and the corresponding changes in host cell biology are interesting, particularly the increase in 2'OH methylation of SARS-CoV-2 genome and a corresponding decrease in host RNA 2'OH methylation. They provide a good explanation for why this may be occurring. Overall, the manuscript is very well written, and I think this will be a great addition to Nat. Comm. Just a few comments follow below.

We thank the reviewer for his positive comments.

Major Comments:

Due to the high level of 2'OH methylation of SARS-CoV-2 RNA, do the authors think this could be affecting SHAPE probing experiments in any way (as the site is no longer likely to be reactive with NAI)? Would DMS probing provide additional information here? An inclusion of this point in the discussion would be nice.

We thank the reviewer for his insightful comments. To address this, we have plotted the distribution of SHAPE-MaP reactivities along 2'-O-methylated sites and non-2'-O-methylated sites in the SARS-CoV-2 genome (Figure 1). Interestingly, we did not observe decreased SHAPE-MaP reactivities along 2'-O-methylation sites. As the extent of RNA modifications at a base can vary (from 0-100%), we hypothesize that only a small fraction of a particular base is 2'-O-methylated in SARS-CoV-2, and hence the remaining proportion of non-2'-O-methylated bases can still react with the SHAPE compound for structure probing. We have included this data in the results section and added it as Supp. Figure 15c in the manuscript.

Figure 1. Violin plots showing the distribution of SHAPE-MaP reactivities along non-2'-O-methylated and 2'-O-methylated sites on the SARS-CoV-2 genome.

Line 233-235: I would expect the overall correlation of structure between the two models to be larger, given it is such a small section of the genome (~1.5%) that is changed. Could more discussion be offered here as to the change in overall structure?

We thank the reviewer for his insightful comments. We have now calculated the correlation of SHAPE-MaP reactivities between WT and $\Delta 382$ for bases with increasing read depth (Figure 2). We observed that bases with high sequencing coverage tend to be correlated better. With bases that have coverages higher than median (50th percentile) having a $R=0.861$. This figure is now our Supp. Figure 11a. We have now included read depth information in our Supp. Table 2 to allow readers to filter according to the depth that they want.

Figure 2. Correlation of SHAPE-MaP reactivities between WT and $\Delta 382$ SARS-CoV-2 genome. The dark dots indicate bases with high read depth.

It's great that the authors applied ScanFold as part of their analytical pipeline. It would be very helpful, however, to include a better description of what was actually done (window size, step size and shuffling protocol) to predict low energy regions. A description of what is meant by low energy region (low z-score?) is also important. The authors could consider referencing two previous applications of ScanFold to SARS-CoV-2:

bioRxiv. 2020 Apr 18;2020.04.17.045161. doi: 10.1101/2020.04.17.045161.

Mol Cell. 2021 Feb 4;81(3):584-598.e5. doi: 10.1016/j.molcel.2020.12.041.

We have now included the parameters of ScanFold in our methods section and have also included the references in our manuscript. The highlighted regions are the lowest 20% of z-scores obtained through ScanFold with the outlined parameters (step size 10 nt, 50 randomizations, window size 120 nt).

Minor comments:

Line 37: "SARS-CoV-2 binds", should say "SARS-CoV-2 genome" so people don't confuse it with the fully formed virus.

We have corrected this in the manuscript.

Lines 131 and 136: " local 'Scan-Fold' energies" this can just be ScanFold
We have corrected this in the manuscript.

Lines 175-179 and 287-288 could use a bit of editing.
We have corrected this in the manuscript.

Line 499: there is a "," after the 1. In the next line they use a "." after the 2. Make these

consistent and perhaps use ")") to better distinguish it?

We have corrected this in the manuscript.

Finally, they refer to the program RNAstructure as "RNA structure" several times, just remove the space.

We have corrected these instances in the manuscript.

-Walter Moss

Reviewer #2 (Remarks to the Author):

In this manuscript, Yang et al applied three different structure-mapping methods to interrogate the RNA structure of the SARS-CoV-2 genome and the previously characterised 382-nucleotides deletion version of the genome, inside Vero cells (monkey origin). The authors identified previously characterised structural elements, report on long-range interactions along the genome and discover interactions between the virus genome and host RNA. Using a long-read sequencing platform, the authors identified some differences between the structure of different subgenomic RNAs. The authors identified binding between the virus genome and a host SnoRNA involved in 2'-O-methylation. Using a technique to map 2'-O-methylation sites on RNA, the authors identified several methylation hotspots along the virus genome.

Overall, the findings are important and well controlled. Some of the structural models are in agreement with previous publications, while others (for example, direct interactions with snoRNAs) are novel and very interesting. I therefore recommend publication in Nature Communications pending minor revisions, below.

We thank the reviewer for his/her positive comments.

1. Several different structural models for the SARS-CoV-2 frameshifting element have been recently suggested, including [doi.org/10.1016/j.molcel.2020.12.041], [doi.org/10.1101/2020.06.29.178343], and [doi.org/10.1074/jbc.AC120.013449]. Wherever possible, the authors should align their structural data to each of the current models and show whether one of the previously suggested conformations is more supported by their data.

We thank the reviewer for his/her insightful comments. We have now mapped our SHAPE-MaP reactivity data for both the WT and \$\Delta\$ 382 viruses onto the structure models from other groups (as suggested by the reviewer above). We observed that our data best agrees with the in cell structure model from the Rouskin lab (Figure 3). We have now included the results in Supp. Figures 3 and 4.

Figure 3. Mapping of our SHAPE-MaP reactivities from the WT SARS-CoV-2 virus genome onto predicted structure models.

There are minor differences between the frameshifting element structures shown in figure 1 and supplementary figure 2. The authors should explain the reason for these differences.

The frameshift element structure model in Figure 1 and Supp. Figure 2 represent the best structure and the best pseudoknot structure from RNAStructure program that is constrained by our SHAPE-MaP reactivities.

2. The manuscript would benefit from adding some background on the function of the frameshifting-element.

We thank the reviewer for his/her comments. We have now included additional background on the function of the frameshifting-element in our manuscript.

3. The authors used their previously developed SPLASH method to analyse RNA-base pairing along the virus genome. It would be important to detail the depth of the data (how many chimeric reads have been identified overall) and the type of control used for these measurements.

We thank the reviewer for his/her comments. The SPLASH numbers can be found in Supp. Table 1 of our manuscript and in Table 1 of the rebuttal.

Table 1. Statistics of SPLASH libraries

SPLASH Statistics	Library ID	Total reads	Unique mapped reads	No. of chimeric reads
Vero cells with SARS-CoV-2 WT rep1	RHH8409.	60980354	12675914	73530
Vero cells with SARS-CoV-2 WT rep2	RHH8412	57868539	12802674	76731
Vero cells with SARS-CoV-2 WT rep3	RHH8457	43334526	9364311	88467
Vero cells with SARS-CoV-2 WT rep4	RHH8460	45844528	10082600	81503
Vero cells with SARS-CoV-2 d382 rep1	RHH8408	62451254	16894818	65106
Vero cells with SARS-CoV-2 d382 rep2	RHH8411	59394244	17391617	84392
Vero cells with SARS-CoV-2 d382 rep3	RHH8458	58750593	8830830	83302
Vero cells with SARS-CoV-2 d382 rep4	RHH8461	53633846	9497734	90310
Vero cells rep1	RHH8410	63569555	13953269	122775
Vero cells rep2	RHH8413	57204450	12875317	115823
Vero cells rep3	RHH8459	59867468	10330294	128456
Vero cells rep4	RHH8462	87098758	14141152	169477

As control, we analysed the rRNA crystal structure to determine the proportion of SPLASH interactions that fall within close physical proximity of less than 30Å. 82.9% of our SPLASH chimeric interactions on 18S and 28S rRNA interactions fall within 30Å, suggesting that our data is of good quality (Figure 4). This figure is now Supp. Figure 7c in the manuscript.

Figure 4. Distribution of the distances captured on the 18S and 28S rRNA crystal structure using SPLASH. Most of the SPLASH chimeras are within 30Å.

4. The authors compare the structure of the wild type and the deletion version of the virus. However the correlation between the SHAPE reactivity of these two genomes (supplementary figure 8a) seems greater than the level of reproducibility between replicates

of the same genome (supp. Figure 1C). The identified differences would therefore benefit from appropriate statistical analysis.

We thank the reviewer for his/her comments. We have now included how we calculated the p-values for WT and $\Delta 382$ reactivity comparisons in the methods section, included the p-values in the figures and figure legends.

5. The comparison between the structure of the virus genome and the different subgenomes would also benefit from statistical analysis. The authors should mention that some structural differences between the SARS-CoV-2 genome and subgenomes were previously identified using a pull-down strategy to separate genomes from subgenomes [doi.org/10.1016/j.molcel.2020.11.004].

We thank the reviewer for his/her comments. We have now included this in our manuscript.

6. The interaction between SNORD27 and the viral genome does not seem to coincide with the identified 2'-O-methylation sites. The authors should refer to that in their discussion.

We thank the reviewer for his/her comments. We have now included this in our discussion.

Reviewer #3 (Remarks to the Author):

In the manuscript “Comprehensive mapping of SARS-CoV-2 interactions in vivo reveals functional virus-host interactions”, the authors use SHAPE mutational mapping, PORE-cupine, SPLASH and Nm-seq to identify novel divergent gRNA structures, compare structure and folding of WT and $\Delta 382$ virus genomes, identify direct virus and host-genome interactions, and explore potential effects of the virus on gene regulation of the host. There are a few instances where the authors do not fully explain their reasoning for or the implications of a finding, and where they overstate, but overall this seems like a novel and significant contribution to the understanding of SARS-CoV-2 infection and interaction with the host, and provides a direction for future studies of viral mechanism.

We thank the review for his/her positive comments.

Comments:

1. Fig 2b: what do these highly structured elements tell us? The use for ssRNA regions that could be targeted is clear, but what can be predicted about the function of the novel structured elements that are displayed here? I'd add that the way the structures for the RNA are plotted makes it difficult to immediately derive meaning - I'd suggest showing on or two specifically interesting regions, along with a clearer annotation of where the siRNA target sites are and what gene they are from.

We thank the reviewer for his/her comments. We have changed the way that we displayed the structures in Figure 2b and annotated them better to indicate which gene they are from. We have also included a table (Supp. Table 3) that contains the location and sequence of conserved single-stranded regions for siRNA targeting along the virus genome.

2. Fig 3a: Is there a way to quantify % of interactions that are the same in WT versus delta382 genomes? It appears robust, but statistics on the correlation of interactions would be helpful. Perhaps an upset plot or venn diagram with a filtered strength of interaction?

We thank the reviewer for his/her comments. We have now plotted a venn diagram and observed that 89% of the interactions in $\Delta 382$ virus are shared with the WT, and 68% of the WT interactions are shared with $\Delta 382$. This data is now included as Supp. Figure 7d,e in the manuscript. For the shared interactions between WT and $\Delta 382$, their read counts are highly

Figure 5. Left, Venn diagram showing the overlap in intramolecular pair-wise RNA-RNA interactions using SPLASH in WT and mutant viruses. Right, Scatterplot showing the correlation in read count for the shared 166 chimeric interactions between WT and mutant viruses.

correlated to each other ($R=0.923$).

3. Fig 3d: ribosome pause sites do not have significantly more SPLASH reads; also what occupancy is included in the “all sites”; could they provide a list? Why are the reads so low compared to the other interaction length data; is it because of the ribosome profiling dataset? Why not compare the pause sites to non-pause sites versus all sites?

We thank the reviewer for his/her comments. We have now provided a list of the locations of all ribosome pause sites as Supp. Table 6. We have now recalculated the SPLASH reads on pause sites versus non-pause sites, in addition to all sites, and showed that the results

Figure 6. Violin plots showing the distribution of SPLASH reads on all sites (left), non-pause sites (middle) and ribosome pause sites (right).

are consistent (Figure 6).

4. There are regions where the SHAPE-MaP and PORE-cupine data do not correlate well (Supp 5d) - any discussion of this? The R^2 is 0.49, which I wouldn't necessarily consider to be a good correlation.

We thank the reviewer for his/her insightful comments. Previous literature has shown that the different high throughput structure probing methods can capture different single-stranded bases along an RNA probably due to different library preparation protocols and sequencing strategies. We have previously benchmarked Pore-cupine with icSHAPE and SHAPE-MaP and showed that the pair-wise overlap between the three methods is around 35% (Figure 7). As such, the SHAPE-MaP and Pore-cupine data do not correlate perfectly, but rather serve

as orthogonal evidence for single-strandedness. However, the Pore-cupine data in itself is highly reproducible (Supp. Figure 8b in manuscript, Figure 8 below), and hence we believe that it provides a robust measure of structure differences between the sgRNAs.

Figure 7. Left, Venn diagrams showing the overlap between the single-stranded bases identified in icSHAPE, PORE-cupine and SHAPE-MaP. Right, boxplots showing the fraction of bases that is identified in PORE-cupine and either SHAPE-MaP or icSHAPE (left), in SHAPE-MaP and either Pore-cupine or icSHAPE (middle), and in icSHAPE and either Pore-cupine or SHAPE-MaP.

Figure 8. Scatterplots showing the correlation between two biological replicates of PORE-cupine reactivity for each subgenomic RNA.

5. Figure 4 is novel in that the authors identify different structure models for sgRNAs, but biological context would be helpful; how does this compare to other viruses - even if predicted, what does this mean for viral-genome interactions that are shown here, why did they want to determine the structures for these sgRNAs?

We thank the reviewer for his/her insightful comments. SARS-CoV-2 uses discontinuous replication to generate a series of subgenomic RNAs that encode important virus proteins such as the envelop, spike, membrane and the nucleocapsid protein. The abundance of these transcripts can vary greatly, such as N's abundance is much higher than the other subgenomic RNA's. However how they are regulated at the level of transcription and translation is still largely unknown. As RNA structures are known to regulate transcription, translation and decay, we were motivated to probe the RNA structures of these subgenomic RNAs to understand the role of structure in gene regulation. We observed a weak

correlation between sgRNA structure and abundance levels, suggesting that structure may play a role in transcript regulation.

6. Can the authors conjecture why the top 10 strongest interactors contributed to slightly more delta382 binding events than WT binding events? Are additional binding events lost in the delta382 strain that would cause the others to be higher contributors?

We thank the reviewer for his/her comments. We did observe fewer number of intermolecular interactions between $\Delta 382$ genome and host RNAs (365 interactions) as compared to WT genome with host RNAs (414 interactions, Figure 9). As such, the top 10 strongest interactors in $\Delta 382$ genome could contribute to more binding events than the top 10 interactors in the WT genome. Additionally, we also observe that the $\Delta 382$ virus grows faster inside the cells than the WT virus. As such, its genome might also have a stronger

Figure 7. Venn diagram showing the number of virus-host interactions for the WT genome and the $\Delta 382$ genome.

interaction with host and the WT genome. This is now our Supp. Figure 13c.

7. I'm curious about the SPLASH interaction between the viral and host RNAs - specifically I'm interested in how normalization/controls were done to account for the proposed strong interaction with mitochondrial RNA. How does this compare to other splash data? Is it normalized to account for the expression level of mito RNAs? Do other exogenously expressed RNAs or other viruses show a similar type of interaction?

We thank the reviewer for his/her comments. We did observe a correlation between virus-host RNA interaction count and host RNA abundance inside the cell. We calculated confidence intervals and identified virus-host RNA interactions that are two standard deviations away from the amount of interactions we would expect based on host RNA abundance. We observed that mitochondrial RNAs, including COX1, COX2, ND5,

Figure 8. Scatterplot showing the correlation between virus-host SPLASH interactions and host cellular abundance for WT (left) and $\Delta 382$ (right) genome. The orange line indicates the best fit line. The grey line indicates the 95% confidence interval of the best fit line. The yellow line indicates 2 standard deviations from the best fit line. Interactions above the yellow have higher SPLASH interactions with virus than would be expected from their abundance.

mitochondrial rRNA and tRNAs show significant interactions with the SARS-CoV-2 genome (Figure 10). We have now included this figure as Supp. Figure 13e in the manuscript.

8. In Fig6, can the authors conjecture why 2'-O-methylation might also be enriched at I codons in addition to the 5' and 3' UTRs?

We thank the reviewer for his/her comments. In order to understand the relationship between 2'-O-methylation sites and I codons on the SARS-CoV-2 genome, we plotted the distribution of I codons and 2'-O-methylation sites. We observed high levels of 2'-O-methylation not only in UTRs, but also towards the end of the genome, in subgenomic RNAs from M-N RNAs. This coincides with an increase in I codons towards the end of the genome, as such we observe an enrichment of 2'-O-methylation sites at I codons (Figure

Figure 9. Probability of the presence of Isoleucine codons (top) and 2'-O-methylation sites (bottom) along the SARS-CoV-2 genome

11).

9. The authors say that the distance from the 2'-O-methylation site are closer to virus-host interaction sites, but by eye it looks pretty similar between uninfected and infected cells, and the p-value is right on the edge of significance....maybe a more interesting way to look at this is to think about the minimum distance overall between the sites of 2'-O-methylation and potential interaction sites, whether or not the cells are infected.

We thank the reviewer for his/her comments. We have calculated the minimum distance between 2'-O-methylation sites and interaction sites before and after SARS-CoV-2 infection and observed a small significant difference in the distance of the 2'-O-methylation sites (Figure 12). We additionally tested whether newly gained 2'-O-methylation sites, upon virus infection, are closer to SPLASH interactions than 2'-O-methylated sites that we found only before virus infection. Again we observed a small but significant difference between them

Figure 10. Left: Violinplots showing the distribution of the distance between 2'OMe site on host RNAs to virus-host RNA SPLASH interaction site, before and after SARS-CoV-2 virus infection. Right: Violin plots showing the distribution of the distance between 2'OMe sites and SPLASH interactions for 2'OMe sites that are found only in uninfected cells, newly gained in infected cells and in both.

(Figure 12).

10. In 6k, though there is an decrease/increase in abundance of RNAs, but static information cannot give information about decay, so they just have to be careful about their claims; perhaps there could be a feedback loop and decrease in production as well.

We agree with the reviewer and have toned down our claims.

11. Finally, given that the SHAPE reagents target the 2'-hydroxyl site as well, I'm curious if the 2'-O' methylation is detectable from the shape-map or PORE-cupine results? Is it blocked or otherwise detectable through a shift in reactivity?

We have now plotted the distribution of SHAPE-MaP reactivities on 2'-O-methylated and non-2'-O-methylated sites along the SARS-CoV-2 genome. We did not observe a decrease in SHAPE-MaP reactivities along 2'-O-methylation sites. As the extent of RNA modifications at a base can vary (from 0-100%), we hypothesize that only a small fraction of a particular base is 2'-O-methylated in SARS-CoV-2, and hence the remaining proportion of non-2'-O-methylated bases can still react with the SHAPE compound for structure probing. We have added this data as Supp. Figure 15c in the manuscript.

Figure 11. Violin plots showing the distribution of SHAPE-MaP reactivities along non-2'-O-methylated and 2'-O-methylated sites on the SARS-CoV-2 genome.

Reviewer comments, second round –

Reviewer #1 (Remarks to the Author):

The authors have addressed all my comments and concerns. I believe this revised manuscript is suitable for publication.

Reviewer #3 (Remarks to the Author):

I'm satisfied with the changes/additions made to the manuscript and find it suitable for publication.